# Machine learning reveals cryptic dialects that explain mate choice in a songbird

Daiping Wang [1,8,11✉], Wolfgang Forstmeier [1,11✉], Damien R. Farine [2,3,9✉],
Adriana A. Maldonado-Chaparro [2,3,4,10], Katrin Martin[1], Yifan Pei [1], Gustavo Alarcón-Nieto [2,5],
James A. Klarevas-Irby[3,4,6,9], Shouwen Ma [7], Lucy M. Aplin [3,5] & Bart Kempenaers [1✉]

Culturally transmitted communication signals – such as human language or bird song – can change over time through cultural drift, and the resulting dialects may consequently enhance the separation of populations. However, the emergence of song dialects has been considered unlikely when songs are highly individual-specific, as in the zebra finch (*Taeniopygia guttata*). Here we show that machine learning can nevertheless distinguish the songs from multiple captive zebra finch populations with remarkable precision, and that 'cryptic song dialects' predict strong assortative mating in this species. We examine mating patterns across three consecutive generations using captive populations that have evolved in isolation for about 100 generations. We cross-fostered eggs within and between these populations and used an automated barcode tracking system to quantify social interactions. We find that females preferentially pair with males whose song resembles that of the females' adolescent peers. Our study shows evidence that in zebra finches, a model species for song learning, individuals are sensitive to differences in song that have hitherto remained unnoticed by researchers.

[1] Department of Behavioural Ecology and Evolutionary Genetics, Max Planck Institute for Ornithology, 82319 Seewiesen, Germany. [2] Department of Collective Behavior, Max Planck Institute of Animal Behavior, 78457 Konstanz, Germany. [3] Center for the Advanced Study of Collective Behaviour, University of Konstanz, Universitätsstrasse 10, 78457 Konstanz, Germany. [4] Department of Biology, University of Konstanz, Universitätsstrasse 10, 78457 Konstanz, Germany. [5] Cognitive and Cultural Ecology Research Group, Max Planck Institute of Animal Behavior, Radolfzell, Germany. [6] Department of Migration, Max Planck Institute of Animal Behavior, Radolfzell, Germany. [7] Department of Behavioural Neurobiology, Max Planck Institute for Ornithology, Eberhard-Gwinner-Straße, 82319 Seewiesen, Germany. [8] Present address: CAS Key Laboratory of Animal Ecology and Conservation Biology, Institute of Zoology, Chinese Academy of Sciences, Beijing, China. [9] Present address: Department of Evolutionary Biology and Environmental Studies, University of Zurich, 8047 Zurich, Switzerland. [10] Present address: Department of Biology, Faculty of Natural Sciences, Universidad del Rosario, Bogotá, D.C., Colombia. [11] These authors contributed equally: Daiping Wang, Wolfgang Forstmeier. ✉email: wangdaiping@ioz.ac.cn; forstmeier@orn.mpg.de; damien.farine@ieu.uzh.ch; b.kempenaers@orn.mpg.de

In many species, including in primates[1], cetaceans[2] and birds[3,4], individuals learn song or contact vocalisations from social interactions with their parents or with other conspecifics[5,6]. From the receiver side, recognition of song is also learnt, typically involving sexual imprinting either on parents or on other members of the population[6–9]. Such culturally inherited traits may be passed on from one generation to the next with imperfect fidelity, leading to divergence between isolated populations via cultural drift[10–14]. Just like the diversification of human languages and dialects[15,16], geographically separated populations of animals with learnt vocalisations (mostly passerine birds) have diverged culturally into geographically restricted song dialects[3,4,17–20]. Cultural conformity within local dialects ensures that the signal will be recognised by receivers[18]. However, conformity may be limited when sexual selection favours greater song complexity for individual males[21,22] or when benefits of signalling individual identity[23] favour greater variability between males. In such cases, the need for individual recognition and distinctiveness may favour individuals that produce innovative or unusual songs[24,25], thereby eliminating local dialects[24–26]. However, in some such species, playback experiments have provided contradictory results, with individuals still able to discriminate between local and non-local song despite no apparent difference in dialect[27–29], i.e., in the measured song parameters[4].

For the zebra finch, the best-studied species in terms of song, the prevailing view is that the large between-individual variation (i.e., the prominent individuality of songs) effectively hinders the emergence of any salient group differences (i.e., between-population divergence)[24,25]. Song learning in zebra finches occurs within a short period during adolescence after which songs are more or less fixed for life (closed-ended learning[30]). Only males sing, and sons mostly learn from their fathers, but also from other tutors, both in captivity and in the wild[31–33]. Since song plays an important role in mate choice[34], it has been proposed that females might prefer songs similar to those they grew up with[7,35]. Yet, only limited geographic variation in song has been found in the wild[36,37]. Extending on earlier work[36–38], a sophisticated and comprehensive study[24] of songs of 12 captive and one wild zebra finch population concluded that population divergence in song was minimal, and hence that 'it seems unlikely that zebra finches would prefer an unfamiliar song from their own population over a song from other populations'. This conclusion was further supported by a simulation[24] showing that distinctive group signatures cannot emerge in species where song learning is not characterised by a bias towards conformity[18], but rather by a high rate of innovation (concerning 15% to 50% of song elements[24,32,39–41]) and an anti-conformity bias to preferentially learn rare rather than common song elements[42,43].

In contrast to this earlier work, we show that zebra finches seem surprisingly sensitive to population differences in song during the process of mate choice, and that a machine learning algorithm can assign individual songs to our four captive populations with only little error, suggesting the existence of 'cryptic song dialects' (i.e., population differences in song that have hitherto remained undetected).

## Results

**Study populations.** We used four captive populations of zebra finches that have been isolated from one another for different amounts of time. These include two domesticated populations ($D_1$ and $D_2$) that have been in captivity for about 100 generations, and two populations ($W_1$ and $W_2$) that came from the wild about 25 and 5 generations ago, respectively (Supplementary Figs. 1 and 2). Due to selective breeding by aviculturists, individuals from the domesticated populations are distinctively larger than more

recently wild-derived birds (about 16 vs 12 grams; Supplementary Table 1, Supplementary Fig. 3). An earlier methodological study[44] reported that when mixing groups of domesticated and wild-derived zebra finches, the previously unfamiliar individuals paired assortatively by population (22 out of 27 pairs, 81%). The authors suggested that this pattern might be due to sexual imprinting 'with individuals preferring to mate with birds that resemble their parents in size and morphology'[44] (i.e., imprinting on morphology). Alternatively, the populations used in that study may have undergone song differentiation via cultural drift and individuals may have mated assortatively for song (i.e., imprinting on song). Hence, it remains to be clarified whether assortment occurred because of variation in morphology or in culture (or both).

**Song classification by machine learning.** First, to assess population differences in song, we trained a freely available sound classifier tool that is based on machine learning[45] (Apple Create ML, Sound Classifier, https://developer.apple.com/machine-learning/create-ml/) with two sets of songs coming from two of our four populations (going through all six pair-wise combinations). The algorithms classified between 93% and 97% of the training songs into the correct population category (Table 1). We then tested the validity of these algorithms on an independent data set consisting of song recordings from the subsequent offspring generation. Classification success varied between 85% and 95%, and lay above 91% for all four pairs of populations that have been separated for roughly 100 generations (Table 1). These results suggest that zebra finch populations can differ distinctively in their song.

**Confirming assortative mating by population.** First, we verified that the previously reported[44] pattern of assortative mating holds also for our domesticated and wild-derived populations. We created four mixed-population groups (replicate 1: two groups containing birds from $D_1$ and $W_1$, replicate 2: two groups containing $D_2$ and $W_2$) of unmated individuals and allowed them to freely pair and build a nest over a 2-week period. Each group was housed in a large indoor aviary and consisted of 36 individuals, with equal numbers of males and females, and equal numbers of domesticated and wild-derived birds. All potential mates were unfamiliar to each other, ensuring that mating patterns cannot be

**Table 1 Classification success of song recordings from four captive zebra finch populations based on a machine-learning algorithm (left) and approximate time of population separation in number of generations (right).**

| Classification success | | | | | Population separation | | | |
|---|---|---|---|---|---|---|---|---|
| | $W_1$ | $W_2$ | $D_1$ | $D_2$ | | $W_1$ | $W_2$ | $D_1$ | $D_2$ |
| $W_1$ | | 0.85 | **0.95** | 0.91 | $W_1$ | | 25 | **100** | 100 |
| $W_2$ | 0.95 | | | 0.92 | **0.91** | $W_2$ | 25 | | 100 | **100** |
| $D_1$ | **0.96** | 0.97 | | 0.91 | $D_1$ | **100** | 100 | | >2* |
| $D_2$ | 0.96 | **0.94** | 0.93 | | $D_2$ | 100 | **100** | >2* | |

*Note that population $D_1$ received a 50% admixture of birds from population $D_2$ two generations before Generation 1 of the present study, and after a longer period of isolation (>30 generations). The admixture event may not have eliminated all population differences. Classification success is the proportion of song recordings that is classified correctly in pair-wise comparisons between populations ($W_1$, $W_2$: recently wild-derived; $D_1$, $D_2$: domesticated). Below the diagonal is the classification success during validation based on the training sample (individuals from Generation 1; 60–64 recordings per population; average length of recording: 6.8 s). Values above the diagonal show the classification success based on the independent testing sample (individuals from Generation 2; 2 × 34–40 recordings per population pair, including only birds that were not cross-fostered between populations, see Methods). The expected random classification success equals 0.50. The matrix on the right shows the putative approximate duration of population separation (in number of generations since common ancestor; see Supplementary Fig. 1). Bold print highlights population pairs used in the cross-fostering study. Measures of song differences between these four populations based on similarity scores from Sound Analysis Pro[27] (SAP, version 2011.10460) are given in Supplementary Table 2.

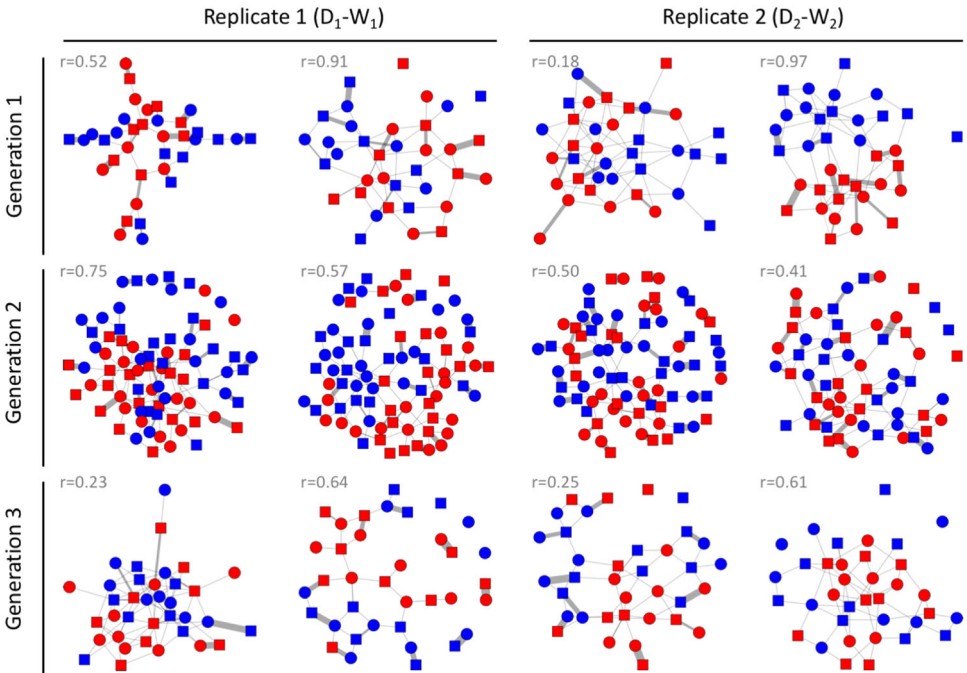

**Fig. 1 Social networks of all experimental groups across three generations.** Each network depicts one aviary with equal numbers of males and females from different backgrounds (as shown in Fig. 3). Symbols (nodes) represent individual males (squares) and females (circles). Lines between the nodes (links) represent the number of associations reflecting pair bonding (allopreening, sitting in bodily contact, and visiting a nest box together). Colours represent the cultural background: domesticated (D, red) and wild-derived (W, blue). The *r*-values are the assortativity coefficients (a network version of the Pearson's correlation coefficient) with regard to cultural background (see Supplementary Table 3 for details). Group sizes are 36 in Generations 1 and 3, and 80 in Generation 2 (except for one aviary with only 32 males and 31 females).

affected by familiarity. Social network analysis of all observations of heterosexual interactions indicating pair formation (allopreening, sitting in bodily contact, and visiting a nest box together) showed that most interactions occurred within the genetic population (Generation 1 in Fig. 1, Supplementary Table 3). The pairings that resulted from those heterosexual interactions showed assortative mating in both replicates (90% and 83% of pairs, respectively; Generation 1 in Fig. 2; Supplementary Table 4). These results confirm strong assortative mating for population of origin[44].

**Hypotheses for causes of assortment**. The observed assortment could be explained by different processes of mate choice and intrasexual competition. Hypothesis 1 assumes an innate preference for a genetic trait (e.g., body size), such that all individuals prefer larger (domesticated) partners. Larger individuals might have priority access to large partners because they are dominant, leaving the non-preferred smaller birds to pair among themselves (i.e., competitive assortative mating by size[46]). Hypothesis 2 assumes a learnt preference for a genetic trait, such that all individuals prefer the morphotype of their foster parents on which they sexually imprinted[44,47,48]. Hypothesis 3 assumes a learnt preference for a cultural trait, such that all birds prefer to mate with a partner from their own cultural population because of socially transmitted variation in song characteristics[6,31].

**Cross-fostering experiment**. To differentiate between these hypotheses, we carried out experiments across two subsequent generations. Birds from each of the four populations (Generation 1) were allowed to breed in large aviaries (each population separately), and we cross-fostered all eggs (soon after laying) within or between populations. This resulted in four types of offspring that differed genetically as well as culturally (see

Generation 2 in Fig. 3), because cross-fostered birds will inherit their morphotype from their genetic parents ('population of origin'), but their song from their foster parents ('population of rearing'). We then placed equal numbers of birds from each of the four cross-fostered types together in indoor aviaries and tested for assortative mating (replicate 1: two groups of $D_1$ - $W_1$, replicate 2: two groups of $D_2$ - $W_2$, each group consisting of 40 males and 40 females, except for one group which only had 32 males and 31 females, see Fig. 1). We used an automated barcode tracking system[49] to capture the process of mate choice in each social group (Supplementary Fig. 4). Every two seconds throughout the day (14.5 h during which the lights were turned on), we identified the nearest male for each female, and constructed a daily social network for each group, reflecting social preferences. After 30 days, we moved each group into a separate, larger outdoor aviary with nest boxes and nesting material and determined which pairs subsequently bred together over a 2-month period.

The three hypotheses make contrasting predictions about which pair bonds should form between the four types of males and females (16 possible combinations; Fig. 4). Birds from Generation 2 showed strong associations (Fig. 1, Supplementary Movies 1–4), positive assortative mating with opposite-sex individuals from their population of rearing (Fig. 2, Supplementary Fig. 5), and strong negative assortment with regard to population of genetic origin (Fig. 2, Supplementary Fig. 5). The observed patterns were highly consistent between replicate 1 (using $D_1$ and $W_1$ birds) and replicate 2 ($D_2$ and $W_2$ birds; Fig. 2 and Fig. 5a, b). These results are clearly incompatible with Hypothesis 1 (innate preference for a genetic trait; e.g., assortative mating by size), and provide little support for Hypothesis 2 (sexual imprinting on the morphotype of the parents). Instead, they fit best Hypothesis 3, i.e., learnt preference for a cultural trait (Fig. 4). This conclusion is strengthened by the observation that assortment by size did not occur within genetic populations

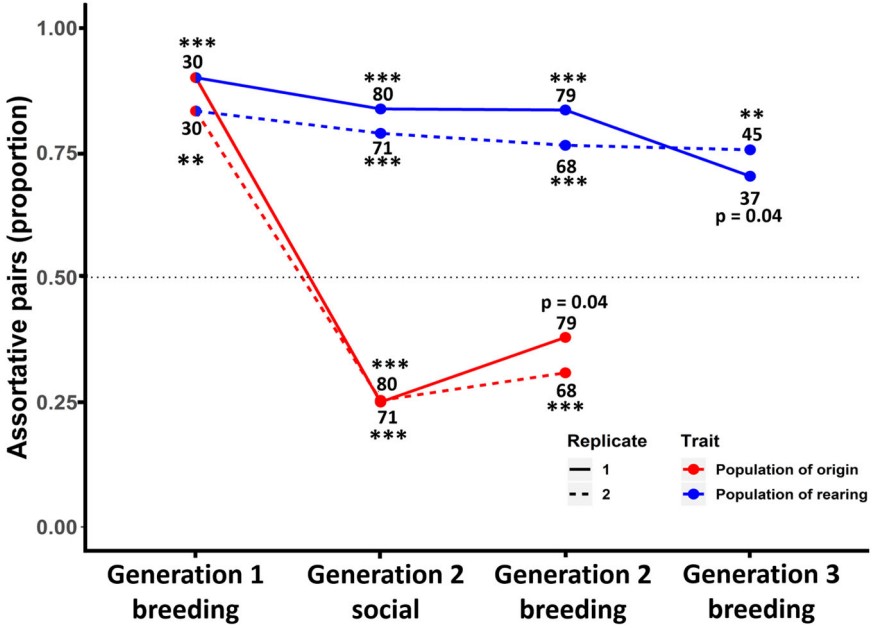

**Fig. 2 Results of tests for assortative mating.** Patterns of assortative mating over three experimental generations (see Supplementary Table 4 for exact *p* values). The *y*-axis shows the proportion of social pairs that were assortative with regard to traits that can only have been culturally transmitted such as song (blue) and traits that have been genetically inherited such as body size (red). The black dotted line marks the random expectation of 50% assortative pairs given an equal number of birds in each category. The two replicates, 1 and 2, are indicated by solid and dashed lines, respectively. The total number of pairs in each of the two replicates are indicated above or below the dots. ***$p < 0.0001$, binomial test, two-tailed; **$p < 0.001$; *$p < 0.01$. In Generation 1, where populations differed culturally and genetically, most individuals paired assortatively by population. In Generation 2, after cross-fostering, individuals mated assortatively by cultural background (population of their rearing parents) and disassortatively by genetic background (population of origin). In Generation 3, where tests were carried out within each genetic background but included groups that differed in cultural background, pairs formed assortatively by cultural as opposed to genetic background.

(Supplementary Fig. 6). Analysis of daily social networks within and between sexes revealed that the patterns of assortment by song and dis-assortment by population of origin occurred only between sexes (Fig. 6a) but not among same-sex individuals (Fig. 6b), and that the patterns gradually emerged and strengthened over the course of the experiment (Fig. 6a). This indicates that the populations were initially well-mixed and remained well-mixed in terms of same-sex relationships, but separated over time due to mate choice. The sex-specificity of the pattern suggests that the population separation was caused by mate choice, rather than by a hypothetical alternative mechanism based on differences in same-sex familiarity.

Although the results are most consistent with Hypothesis 3 ($r = 0.63$; Fig. 4), there is still more unexplained variance than expected from measurement error alone (note the high repeatability between replicate 1 and 2: $r = 0.92$; Fig. 5b). Thus, in Supplementary Note 1, we consider and discuss *post-hoc* explanations that describe the observed data best (Supplementary Fig. 7). Briefly, the best-fitting explanation is one where assortative mating by song plays the predominant role, but with an additional effect of imprinting on parental morphotype and a tendency for wild-derived birds to prefer (genetically) domesticated birds.

**Analysis of pair-wise distances**. In the preceding analysis, we used categorical predictors (e.g., same rearing environment or not) to explain categorical outcomes (paired or not). However, with such an approach, it remains unclear whether the assortative mating by rearing environment is due to assortment by song dialect or by other culturally inherited traits linked to the rearing environment. Therefore, we next analysed the extent to which individual-specific phenotypes (on a continuous scale) can

explain the variation in male-female social behaviour (in terms of pair-wise proximity) during the 30 days of automated tracking ($n = 5561$ male-female combinations with complete data). As continuous predictors, we fitted (1) the difference in body size between a male and a female, (2) the similarity of the male's song to songs from the female's rearing aviary, as quantified by Sound Analysis Pro[50], and (3) the corresponding song similarity measure, as quantified by the machine learning tool (the latter two predictors are only weakly positively correlated; $r = 0.17$, $n = 584$; Fig. 7). These continuous predictors were examined in combination with the categorical predictors, which are not based on individual characteristics but treat all male-female combinations from one of the 16 pairing categories in the same way.

A first model without the individual-specific predictors confirmed the previous results, i.e., assortative mating by rearing environment (in line with imprinting on dialect), an effect of imprinting on parental morphotype, and a tendency for wild-derived birds to prefer domesticated ones (see Supplementary Table 5). Adding the individual-specific predictors confirms that body size per se has no explanatory power. However, spatial proximity between males and females is predicted by the similarity of a male's song to the songs of the individuals with whom the female grew up. More specifically, it was the similarity to the songs of the peers in her rearing aviary, and not the similarity to the songs of the adult males that bred in the female's rearing aviary (the parental generation 1; Table 2). Intriguingly, both methods of assessing song similarity independently support the conclusion of song-imprinting on peers rather than fathers (Table 2). Even after accounting for rearing environment as a category, both measures of song similarity to the female's peers are significant predictors (Table 2), presumably capturing different aspects of song similarity. Note that the binary predictor of rearing environment ('Imprinting on song' in Table 2)

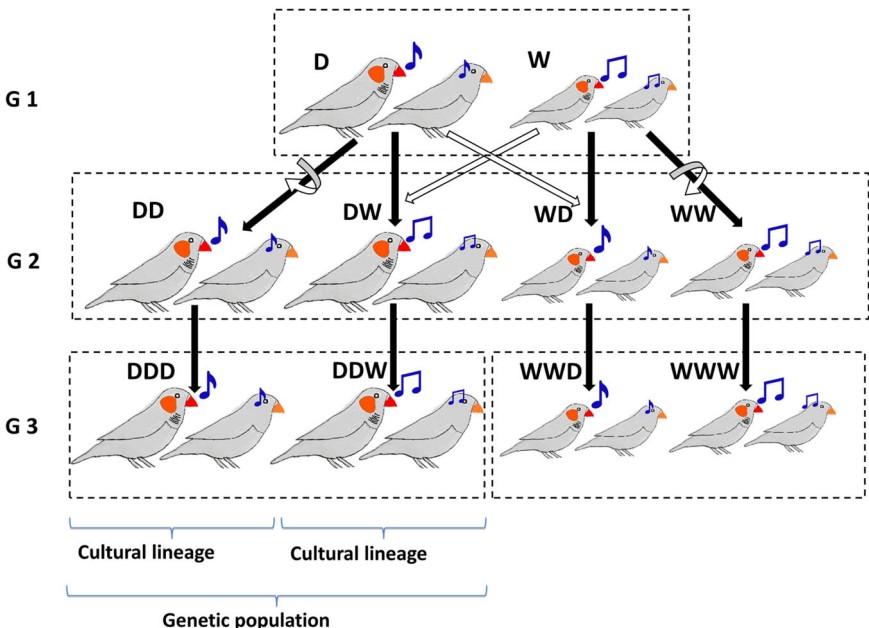

**Fig. 3 Schematic representation of the experimental groups across three generations.** In Generation 1, assortative mating was tested in groups (indicated by the dashed rectangle) consisting of birds from two populations (one domesticated, D, and one wild-derived, W) that differ genetically (e.g., in body size, indicated by the size of the birds) and culturally (e.g., in song, indicated by the shape of the notes). After testing, the two populations were housed separately and four lineages were created by cross-fostering (solid arrows reflect genetic descent, open arrows indicate rearing parents, whereby the curved and straight arrows reflect the within- and between-population cross-fostering, respectively). These four lineages (Generation 2) are denoted as DD, DW, WD, WW; the first letter indicates the genetic population of origin and the second indicates the population of the rearing parents. In Generation 2 assortative mating was tested in groups that contained equal numbers of all four types of males and females. After testing, the four lineages were again housed separately and bred without cross-fostering, such that they passed on their culturally acquired traits to Generation 3. In this generation, assortative mating was tested in groups of males and females with a similar genetic background, but that differed in the cultural traits transmitted through the foster grandparents (indicated by the third letter; e.g., DDW corresponds to birds with genetic background D, raised by parents DW from Generation 2). All experiments were performed with two domestic and two wild-derived populations (replicate 1: $D_1$-$W_1$, replicate 2: $D_2$-$W_2$).

statistically controls for all culturally inherited traits that may differ between the populations and that may be confounded with dialects. Thus, statistically controlling for group differences allows us to focus on variance in traits within populations. Another main strength of this approach is that the two significant individual-level predictors of song similarity cannot easily be confounded by other cultural traits. This is because the birds whose songs are being compared (focal vs peers) have never met and were not reared by the same parents. Hence, these birds cannot have co-inherited other cultural traits in proportion to song similarity, unless they were part of the song system. Instead, the variance in song similarity is idiosyncratic. These results, therefore, support the hypothesis that song similarity to the female's rearing environment is the predominant factor underlying female mate choice. Yet, strictly speaking, the evidence that song per se affects mate choice is correlational rather than fully experimental. Although full experimental control over the songs of zebra finches cannot be achieved, specific breeding designs allow further hypothesis testing. Thus, we produced a third generation of birds to specifically test for the trans-generational effects of song culture within genetic populations (Generation 3 in Fig. 3).

**Maintaining dialects within genetic populations.** Song learning in male zebra finches occurs within a short period during adolescence[30]. This implies that the cross-fostered birds from Generation 2 had acquired their songs from their foster fathers (Generation 1), and passed on these songs to their offspring (Generation 3). Thus, if variation in song is the underlying cause, the mating behaviour of Generation 3 individuals should still be

explained by the original population of rearing (via the effect of the foster grandparents from Generation 1 on the song of the Generation 2 fathers). In contrast, if Generation 2 had acquired other behavioural traits relevant for mate choice while interacting with birds from other populations during the 3-month period they spent together (and assuming open-ended learning for these traits), and passed these behaviours to their offspring, we predict no or little influence of the foster grandparents (Generation 1) on the mating behaviour of individuals from Generation 3. To test these alternatives, we mixed birds from the two cultural lineages that had been established within each genetic population (see Fig. 3). Thus, in this experiment, effects of morphological differences between populations are excluded, because the tests were done within genetic populations.

Our results show that individuals from Generation 3 mated assortatively according to the culture of their foster grandparents in Generation 1 (Fig. 1, Fig. 2, Supplementary Tables 3, 4), while pairings were again random with respect to body size variation within each genetic population (Supplementary Fig. 6). These results further support the hypothesis that mate choice targets cultural traits that are transmitted during a short developmental time window.

## Discussion
**Ruling out confounding factors.** Even though males also learn their 'distance call'[51] and their courtship dance[52] from adult tutors, results from previous studies suggest that it is unlikely that these other cultural traits confound our findings on the role of song. First, the traits of a male's distance call (including voice characteristics) correlated only weakly with the corresponding

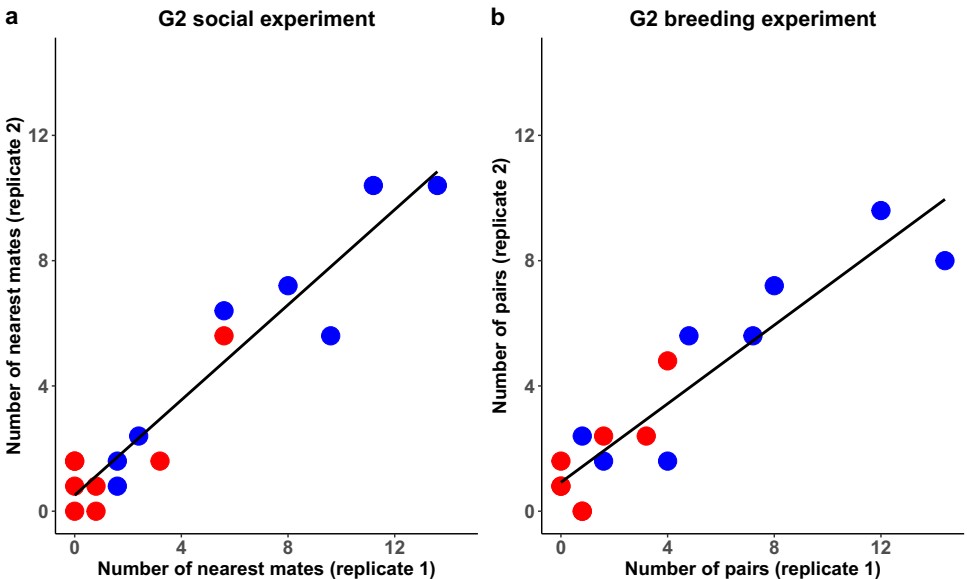

**Fig. 4 Expected versus observed mating patterns in the cross-fostered Generation 2.** The first column indicates three a priori hypotheses (1, 2, 3) and the observed mating pattern ($N = 147$ nesting pairs). The second column shows mating patterns between four types of females (top) and males (bottom): DD, DW, WD, WW (the first letter indicates the genetic population of origin, the second letter indicates the population of the rearing parents; see Fig. 3). The thickness of the blue lines corresponds to the numbers of expected or observed pairs of each male-female combination. The smaller font size for wild-derived birds illustrates their smaller body size. The third and fourth columns show the expected or observed overall correlation between partners with regard to their size category (large D or small W) and song type (D or W, as learnt from foster parents). The last column shows the Pearson correlation coefficient between expected and observed numbers of pairs across the 16 pair combinations. See also Supplementary Fig. 7 for *post-hoc* combinations of multiple hypotheses explaining the observed data.

**a**  G2 social experiment          **b**  G2 breeding experiment

**Fig. 5 Repeatability of pairing behaviour between replicates.** Shown are the number of associations for each of the 16 possible pair categories (each dot refers to one category, e.g., DD-DW, see Figs. 3 and 4). Blue dots refer to pair combinations that share the same song dialect, while red dots represent disassortative pairings with regard to song. **a** Pairings defined as the nearest individual of the opposite sex (distances averaged across 118 million observations over a period of 30 days, $n = 151$ pairs) in replicates 1 versus 2 (Pearson $r = 0.95$, $p = 1.3 \times 10^{-8}$, two-tailed) in the social experiment with barcode tracking but no nesting opportunities. **b** Observed pairs during the breeding experiment ($n = 147$ pairs) in replicate 1 versus 2 ($r = 0.92$, $p = 4.7 \times 10^{-7}$, two-tailed).

traits of the same individual's song (for 16 traits: mean $r = 0.16$, range: $r = 0.04 - 0.24$, $n = 413$ males)[51]. Second, courtship dance movements are linked to song elements only in a probabilistic way (i.e., the probability of a movement occurring in a given 33 ms time window of a male's song rarely exceeds 30%)[52].

Hence, it seems highly unlikely that the findings shown in Table 2 arose from correlations with these other culturally transmitted traits, especially because the birds that are being compared have never met and have no shared foster parents. If such confounds would exist, they would need to be tightly linked to song to

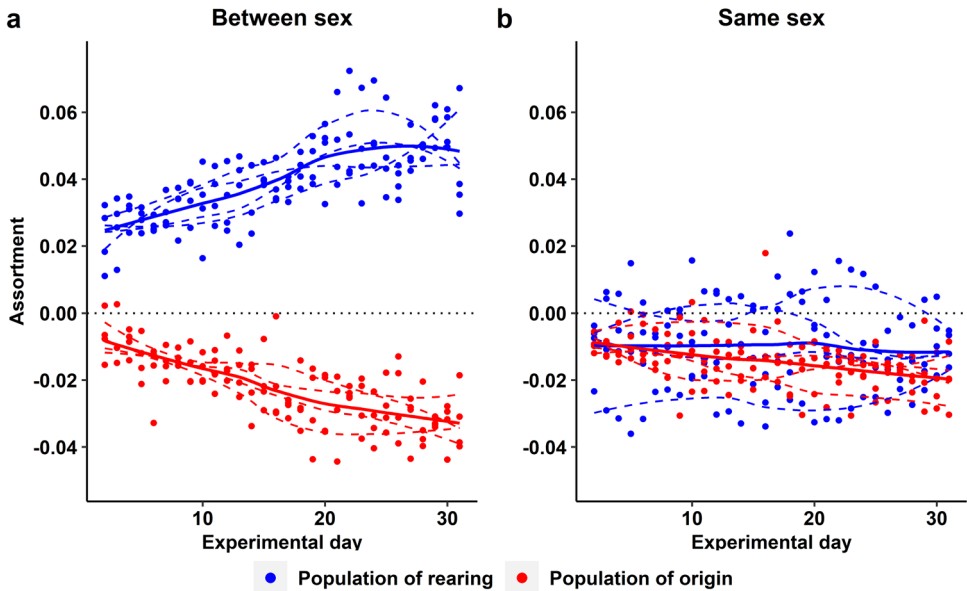

**Fig. 6 Temporal changes in level of assortment between individuals from the same and opposite sex in Generation 2.** Daily values of assortativity coefficients with regard to population of rearing (cultural background, blue) and population of origin (genetic background, red) in each of the four replicate groups. Coefficients are calculated using the distances between all male-female pairs (between sex, (**a**)) or using the distances between all male-male and all female-female pairs (same-sex, (**b**)). Positive and negative coefficient values indicate assortative and disassortative association, respectively. Dashed lines are fitted to each of the four groups separately; the bold lines indicate the fit to the entire data set. Note how in (**a**) heterosexual relationships (based on proximity) progressively become more assortative for the cultural background and more disassortative for the genetic background, while same-sex relationships show no clear deviation from randomness.

induce the three effects shown in Table 2 ('Imprinting on song', 'ML similarity to peers', 'SAP similarity to peers'). In such case, it might still be appropriate to label them 'aspects of song dialects'. Further note that song is used by the males specifically in the context of courting females, with about one in four courtships culminating in a copulation attempt (communal breeding aviary data from[53]). Females typically respond to the courtship song with either rejection or engagement in mutual courtship. Thus, it is likely that song plays a key role in female mate choice. In contrast, other vocalisations, including the learnt 'distance call' are not clearly linked to mating behaviour.

We found that sexual imprinting on the morphotype of the parents explained additional variance in the complex but highly repeatable mating patterns of cross-fostered birds (Fig. 4, Fig. 5, Table 2, Supplementary Note 1 and Supplementary Fig. 7). Thus, sexual imprinting on song was not the only mechanism that played a role in mate choice after bringing together discrete populations. As we found no evidence that body size per se was important, we suggest that differences in morphotype not quantified in this study (e.g., shape of the beak)[48] may instead be salient features. Future studies could again use machine learning to capture multidimensional variation in shapes (morphotypes), which presumably differ between domesticated and wild-derived birds[54].

**The role of song dialects.** Our study suggests that population-specific song dialects drive strong assortative mating in zebra finches. Previous work on birds with unambiguous song dialects, i.e., clear geographical transitions in vocal parameters[4], already showed the importance of such dialects for mate choice[4,9,55]. However, our results contradict the view that each zebra finch population already 'used up' most of the possibilities of singing within the limits of innate constraints[24,25,56] due to a high propensity to innovate[24,25,32,39–41] and to preferentially learn rare rather than common song elements[42,43]. Instead, we reveal – using a machine-learning technique – that zebra finch

populations do differ in song (Fig. 7a, b), and we show experimentally that these 'cryptic song dialects' likely have consequences for social behaviour.

Only a minority of bird species with song learning show obvious dialects[4]. The vast majority of species with complex songs[4,57] do not exhibit sharp geographical transitions in vocal parameters – the defining, but also disputed, criterion of what constitutes 'song dialects'[4]. Studies on species with more complex song at best suggested that some hitherto unquantifiable aspects of gradual geographical change may be salient to the birds[28,29,58], or alternatively, that song may have evolved to signal male identity[59,60] and hence may contain no information about group or population. In contrast to the present understanding, our results suggest that population differences in song are highly salient to the birds. Hence, we coin the term 'cryptic dialects', to define dialects that have not been and perhaps cannot be revealed with conventional methods (see Fig. 7 and[24] for additional approaches, all suggesting little population divergence). Our use of the word 'dialect' does not imply that they must be characterised by diagnostic population-specific signatures, but rather that they can be distinguished with little error (Fig. 7a, b). Familiarity with the songs experienced in the natal environment might be a parsimonious and sufficient explanation for the observed heterosexual assortment by natal dialect and for female preferences for males whose song resembles the songs of the female's peer environment.

While behavioural assays that test the discrimination ability of the respective animals are the most informative about the salience of signals, such assays are laborious[35] and sometimes unfeasible in practice. As an alternative, the potential for discrimination based on song has been evaluated by quantifying differences in arbitrary song traits. We suggest that a machine-learning approach can be particularly useful in this context. Although it also cannot replace a behavioural assay, such an approach has several advantages: it is arguably (1) more sensitive, (2) closer to an individual brain that distinguishes familiar from unfamiliar,

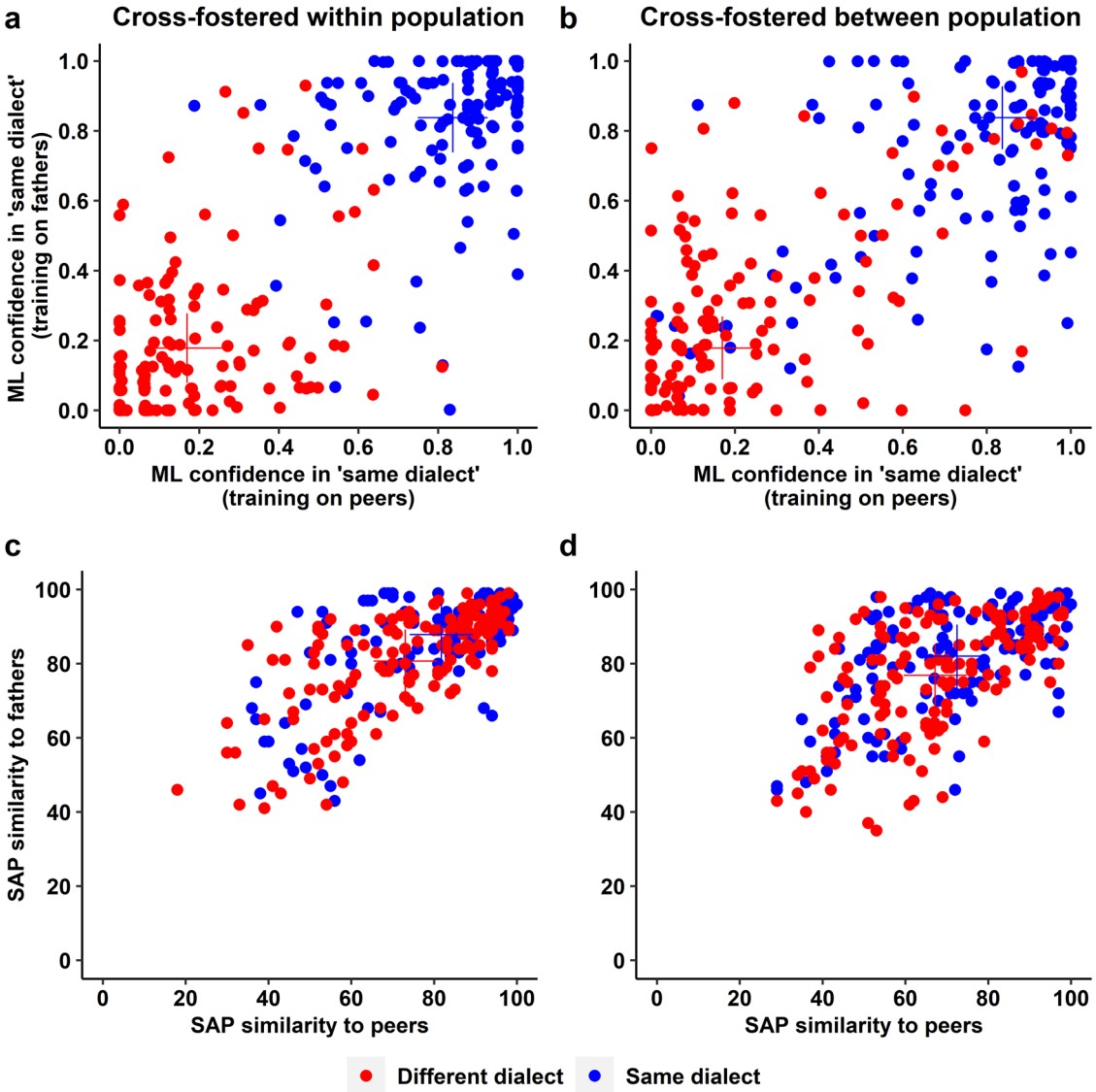

**Fig. 7 Classification scores from a machine-learning algorithm (ML; a, b) and similarity scores from Sound Analysis Pro (SAP; c, d). a, b** A machine-learning algorithm was trained on independent sets of zebra finch song recordings to discriminate between 'same' and 'different' dialect from the perspective of an individual female in Generation 2 given her experiences in a rearing aviary. In the training data set 'same' is represented either by the songs of the set of 8 fathers (Generation 1) or the set of 10 peer members (Generation 2) in the rearing aviary; 'different' is represented by the respective songs from an aviary of another population type (domestic D or wild-derived W, by males that will not be encountered in the social or breeding experiment). The 40 males that a female will encounter in the social and breeding experiment (20 of the same song dialect, shown in blue; 20 of a different song dialect, in red) are then classified by ML as either 'same' or 'different' with complementary confidence scores that add up to one. Note that each male contributes 4 data points (2 'same' and 2 'different') because he encounters four types of females (DD, DW, WD, WW) from different rearing aviaries. **c, d** Similarity scores from SAP using the same representation as in (**a**) and (**b**) (similarity to the songs of the peers or fathers of a female's rearing aviary, which the focal male never met, such that any similarity is indirect). The machine-learning algorithm (**a, b**) achieves much clearer differentiation compared to the traditional SAP software (**c, d**). Males that were cross-fostered within population (DD or WW; **a, c**) are discriminated with slightly higher confidence than DW or WD males (**b, d**; see the crosses that mark the group means).

and (3) less arbitrary than the conventional approach of quantifying some measurable characteristics of the signal. The main downside of a machine-learning approach is that it does not provide any information on the underlying properties of the song that differ and allow discrimination. Moreover, it is essential to ensure that background noise in the recordings is not confounded with population differences in song (see Methods).

It remains unclear why differences in song evoke such strong responses in the mate choice of zebra finches, reminiscent of language and cultural barriers in humans[61]. Preferences for natal dialects may arise as a by-product of mechanisms for species recognition[62,63]. Alternatively, female preferences for the local

song dialect may help targeting males with knowledge of the local environment. Further work is needed to determine whether these preferences are fixed, how common such cryptic dialects are in passerines and whether they can lead to reproductive isolation and play a role in speciation.

## Methods

**Ethics oversight**. This study was carried out within the frame of our housing and breeding permit (311.4-si) granted by the Landratsamt Starnberg, Germany. Attachment of backpacks was approved by the Regierung von Oberbayern, Germany (ROB-55-2-2532. Vet_02-17-211).

**Table 2 Mixed-effect model explaining variation in daily distances between all possible male-female pairs across four aviaries with automated tracking of individuals.**

| | N | Estimate | 95% CI | | df | t | p (2-tailed) |
|---|---|---|---|---|---|---|---|
| | | | Lower | Upper | | | |
| Random effects (% variance explained) | | | | | | | |
| Pair identity | 5561 | 41.1% | | | | | |
| Male identity | 146 | 2.7% | | | | | |
| Female identity | 151 | 2.7% | | | | | |
| Pair rearing aviaries | 64 | 1.7% | | | | | |
| Residual | | 51.7% | | | | | |
| Fixed effects | | | | | | | |
| Intercept | | −0.007 | | | | | |
| W prefer D (H1W) | | −0.057 | −0.093 | -0.022 | 43 | −3.1 | 0.003 |
| Imprinting on morphotype (H2) | | −0.068 | −0.104 | -0.033 | 44 | −3.7 | 0.0005 |
| Imprinting on song (H3) | | −0.068 | −0.113 | -0.023 | 101 | −3.0 | 0.004 |
| ML song similarity to peers | | −0.044 | −0.077 | -0.011 | 4376 | −2.6 | 0.008 |
| SAP song similarity to peers | | −0.044 | −0.073 | -0.014 | 438 | −2.9 | 0.004 |
| Excluded fixed effects | | | | | | | |
| Male-female difference in body size | | 0.004 | −0.024 | 0.032 | 3542 | 0.3 | 0.78 |
| ML song similarity to parents | | 0.033 | −0.006 | 0.071 | 4086 | 1.6 | 0.10 |
| SAP song similarity to parents | | −0.015 | −0.047 | 0.017 | 1683 | −0.9 | 0.36 |
| SAP song similarity to foster father | | 0.011 | −0.010 | 0.033 | 3979 | 1.0 | 0.30 |

Daily mean distance (in mm, ln-transformed) of each female-male combination was used as the response variable (N = 165422). As random effects, we fitted male and female identity, pair identity, and the combination of the identities of the female's and the male's rearing aviaries (Pair rearing aviaries). The fixed effect predictors H1W, H2 and H3 are based on the best supported hypothesis in Supplementary Fig. 7 (see legend of Supplementary Table 5 for a detailed explanation of these predictors). The other two covariates are measures of the similarity of the song of a given male to the songs of the males with whom the focal female grew up (peer group members in the female's rearing aviary), one assessed by a machine-learning algorithm (ML, confidence of belonging to the same dialect as sung in the female's rearing aviary), the other by Sound Analysis Pro (SAP, using the values illustrated on the x-axes of Fig. 7). Non-significant, excluded predictors are the difference between male and female body size (see Supplementary Fig. 6), and song similarities to the set of eight parental males in a female's rearing aviary (y-axes of Fig. 7) and to the song of a female's foster father. The negative sign of the included fixed effect estimates reflects greater proximity (smaller distance) to males whose song resembles those of a female's peer group and who fulfil the categorical criteria (e.g., male matches the morphotype that the female imprinted on) as illustrated in Fig. 4. Note that the predictors 'Imprinting on song (H3)' and 'ML song similarity to peers' are strongly correlated (r = 0.81; see Fig. 7). If one of those two predictors is taken out, the other one takes up most of its effect. The three excluded song parameters show the following correlations with included parameters: $ML_{parents}$-$ML_{peers}$ r = 0.82, $SAP_{parents}$-$SAP_{peers}$ r = 0.68, $SAP_{fosterfather}$-$SAP_{peers}$ r = 0.24. Despite the high correlation, $ML_{parents}$ is not a significant predictor (p = 0.87) if included instead of $ML_{peers}$.

**Study populations**. We used four zebra finch populations that are genetically differentiated due to founder effects and selection (see Supplementary Fig. 1 & Fig. 2): two domesticated populations ($D_1$ and $D_2$) that have been maintained in captivity in Europe for about 150 years and two populations ($W_1$ and $W_2$) that have been taken from the wild about 10–30 years ago (see Supplementary Fig. 1). We ran all experiments in two independent replicates. We used individuals from populations $D_1$ and $W_1$ for replicate 1 and individuals from $D_2$ and $W_2$ for replicate 2.

**Breeding experiment Generation 1**. We created four groups of 36 individuals (9 males and 9 females from both a domesticated and a wild-derived population, two groups within each replicate) and put each group separately in an indoor aviary (5 m × 2.0 m × 2.5 m). All individuals had been reared normally by their genetic parents in similar breeding aviaries, were inexperienced (never mated before) and unfamiliar to all opposite-sex individuals. In replicate 1 ($W_1$ – $D_1$, starting December 2016), birds were 142 ± 32 days old at the start of the experiment (range: 101–191 days); in replicate 2 ($W_2$ – $D_2$, starting March 2017), birds were 241 ± 47 days old (range: 151–306 days). In each aviary, we provided nest material and nest boxes to stimulate breeding and observed pair-bonding behaviour for ca. 60 h spread over 14 days. Two observers recorded all instances of allopreening, sitting in bodily contact, and visiting a nest box together, which reflects pair bonding[64].

In total, we observed 3166 instances of heterosexual association among the 4 × 36 individuals (Supplementary Table 3). We defined a pair-bond between two opposite-sex individuals if they were recorded in pair-bonding behaviour at least five times (mean: 22 ± 14 SD, range: 5 – 73). This cut-off was chosen (blind to the outcome of data analysis) based on the frequency distribution showing a clear deviation from a random, zero-truncated Poisson distribution (Supplementary Fig. 8). Using this definition, we identified a total of 60 pairs (30 in each replicate). Of all females, 48 and 6 had a pair-bond with one and two males, respectively (18 females remained unpaired). Conversely, 34, 10, and 2 males had a pair-bond with one, two, and three females, respectively (26 males remained unpaired).

**Cross-fostering for Generation 2 experiments**. After the breeding experiment of Generation 1, in 2017, we established two different cultural lineages within each genetic population by cross-fostering eggs, either within or between populations (Fig. 3). For this purpose, we used 16 aviaries (four per population), each containing 8 males and 8 females of the same population (Generation 1). Individuals were allowed to freely form pairs and breed. We reciprocally exchanged eggs shortly after laying between two aviaries per population (within-population cross-fostering) and between pairs of aviaries from different populations (between-population cross-fostering). This resulted in four cultural lineages per replicate (DD, DW, WD, and WW; Fig. 3). Each lineage was maintained in two separate breeding aviaries to ensure the availability of unfamiliar opposite-sex Generation 2 individuals from the same line. Offspring remained with their foster parents until they reached sexual maturity, when the following experiment started.

**Social experiment Generation 2**. Between December 2017 and March 2018, we put four groups of individuals (two groups for each replicate) in indoor aviaries (same as in Generation 1 experiment). Each group consisted of 10 males and 10 females from each of the cross-fostered groups DD, WW, DW and WD, i.e., a total of 80 birds per aviary, except that one aviary of replicate 2 only consisted of 63 individuals (7DD, 8WW, 8DW and 8WD) due to a shortage of birds. In replicate 1 ($W_1$ – $D_1$, starting December 2017), birds were 170 ± 25 days old at the start of the experiment (range: 105–199 days); in replicate 2 ($W_2$ – $D_2$, starting January 2018), birds were 200 ± 29 days old (range: 120–241 days). We recorded the position of individuals using an automated barcode-based tracking system[31]. We fitted each individual with a unique machine-readable barcode (Supplementary Fig. 4a) and placed eight cameras (8-megapixel Camera Module V2; RS Components Ltd and Allied Electronics Inc.), each connected to a Raspberry Pi (Raspberry Pi 3 Model Bs; Raspberry Pi Foundation) in each aviary. For 30 consecutive days, the cameras recorded individuals at six perches and at two feeders (Supplementary Fig. 4b, c). Between 05:30 and 20:00, when lights were switched on, each camera took a picture every two seconds.

Each day, pictures stored on the Raspberry Pis were downloaded to a central server and processed using customised scripts. The customised software used the PinPoint library in Python[65] to identify each barcode in each picture, allowing us to simultaneously track the position and orientation of each individual (Supplementary Fig. 4b) for the duration of the experiment. The tracking system generated 118 million observations across all four aviaries (Supplementary Fig. 4c). From these data, we extracted the average distance between the male and the female (in mm) for each male-female dyad, either daily or across the entire 30-day period (for comparison, such distance data were also extracted for all male-male and all female-female dyads). We used this dataset to identify the nearest opposite-sex individual for each of 151 males and females (55% of these 151 associations were reciprocal). Out of 151 nearest males to females, 74 (49%) paired with that female in the following breeding experiment (see below) and this proportion strongly increased as the average distance between partners decreased (Supplementary Fig. 9).

**Breeding experiment Generation 2**. Immediately after the social experiment, we moved each group into a separate semi-outdoor aviary (5 m × 2.5 m × 2.5 m) and provided nest material and nest boxes. During the next 2 months, three observers scored heterosexual associations to identify pair bonds as described for 'breeding experiment Generation 1' (ca 300 h per replicate). In total, we observed 6072 associations involving 284 individuals (Supplementary Table 3). Consistent with the previous experiment, we defined a pair-bond when a male-female dyad was observed in pair-bonding behaviour at least five times during the entire experiment (mean: $18 \pm 13$ SD range: 5 - 61; Supplementary Fig. 8). Using this definition, we identified 147 pairs (79 pairs in replicate 1 and 68 in replicate 2). Of all males, 97, 22 and 2 had a pair-bond with 1, 2 and 3 females, respectively (27 males remained unpaired). Conversely, 99, 21 and 2 females had a pair-bond with 1, 2 and 3 males (26 females remained unpaired).

**Breeding experiment Generation 3**. Between April and December 2018, we housed the four cultural lineages (DD, WW, DW and WD) separately again. We placed 8 males and 8 females in each of 16 breeding aviaries (four per lineage) and allowed them to freely form pairs and breed. The offspring belong to four lineages (Fig. 3): two lineages with individuals that had not been cross-fostered between the domestic and wild-derived population (DDD and WWW) and two lineages with individuals from the same genetic background, but where their parents had been cross-fostered and raised by the other population (DDW and WWD).

Between December 2018 and February 2019, we put four groups of 36 birds (two per replicate, i.e., 2 with 18 DDD and 18 DDW individuals and 2 with 18 WWW and 18 WWD individuals; 9 males and 9 females per lineage; Supplementary Table 3) in an outdoor aviary (same as above). In replicate 1 ($W_1 - D_1$, starting December 2018), birds were $172 \pm 44$ days old at the start of the experiment (range: 131–195 days); in replicate 2 ($W_2 - D_2$, starting January 2019), birds were $191 \pm 40$ days old (range: 122–230 days). During 14 days, two observers recorded all pair-bond behaviours as described under 'breeding experiment Generation 1'. In total, we observed 3378 instances of pair-bond behaviour involving 137 individuals (Supplementary Table 3). As above, we defined a pair-bond when a male-female dyad was observed in pair-bonding behaviour at least five times during the entire experiment (mean: $18 \pm 11$ SD, range: 5 - 47; Supplementary Fig. 8). We identified 82 pair bonds (37 in replicate 1 and 45 in replicate 2). Of all males, 34, 16, 4 and 1 had a pair-bond with 1, 2, 3 and 4 females, respectively (17 males remained unpaired). Conversely, 42, 16, 1 and 1 females had a pair-bond with 1, 2, 3 and 5 males, respectively (12 females remained unpaired).

**Morphological measurements**. After birds had reached sexual maturity (>100 days of age), we measured body mass (to the nearest 0.1 g), tarsus length (to the nearest 0.1 mm), and wing length (to the nearest 0.5 mm) of all individuals (all measured by WF). We included these three variables in a principal component analysis (PCA) and used the first principal component (PC1, 67% of variation explained) as a measure of body size.

**Song recording and analysis approach**. We recorded the songs of the parental males from Generation 1 (16 aviaries x 8 males = 128 males, of which 122 were successfully recorded between November and December in 2017) and of their offspring (Generation 2; 146 out of 152 males were successfully recorded between March and May 2018). To elicit courtship song, each male was placed together with an unfamiliar female in a metal wire cage (50 cm × 30 cm × 40 cm) equipped with three perches and containing food and water. The cage was placed within one of two identical sound-attenuated chambers. We mounted a Behringer condenser microphone (TC20, Earthworks, USA) at a 45° angle between the ceiling and the side wall of the chamber, such that the distance to each perch was approximately 35 cm. The microphone was connected to a PR8E amplifier (SM Pro Audio, Melbourne, Australia) from which we recorded directly through a M-Audio Delta 44 sound card (AVID Technology GmbH, Hallbergmoos, Germany) onto the hard drive of a computer.

Previous studies that quantified differentiation of songs between zebra finch populations using specific song parameters (e.g., duration and frequency measures) largely failed to detect prominent differences[12,49,50]. We, therefore, used the following two approaches (Sound Analysis Pro and Machine Learning) to quantify the extent to which a given male's song resembled the songs of other males.

**Song similarity analysis with SAP**. Using Sound Analysis Pro (SAP) version 2011.104[27], we quantified song similarity (ranging from 0 to 100) by direct pairwise comparison of song motifs (the main part of a male's song that is stereotypically repeated and about 0.8 s long, excluding introductory syllables). Pair-wise comparisons of two males (based on one representative motif recording per male) revealed higher within-population similarity than between-population similarity (Supplementary Table 2, data from Generation 1). Further, for offspring that were cross-fostered between populations ($N = 73$ males from Generation 2) song similarity to their foster father was higher than song similarity to their genetic father (80 versus 68, paired $t$-test: $p < 0.0001$). For each of the 146 recorded males of Generation 2, we calculated three measures of song similarity with regard to each of the females encountered in the social experiment with automated tracking of

birds. (1) 'SAP song similarity to foster father': the pairwise similarity between the motif of the focal male and the motif of the foster father of the focal female. (2) 'SAP song similarity to parents': we first combined the song motifs of all eight parental males that were present in the female's rearing aviary (Generation 1) into a single 'super-motif' (simply placing all recordings into a single sound file) and then calculated the similarity of the motif of the focal male to this super-motif from the female's rearing aviary. (3) 'SAP song similarity to peers': we combined the song motifs of all 7-10 recorded peer males present in the female's rearing aviary (Generation 2) into a single 'super-motif' and calculated the similarity of the motif of a focal male to this super-motif.

**Song categorization based on machine learning**. We used the Sound Classifier tool in Apple Create ML (https://developer.apple.com/machine-learning/create-ml/; Version 1.0; 16019; Apple Inc. 2019) in an Xcode environment (Version 11.7; 11E801a) to (1) assess the proportion of individual song recordings that can be correctly assigned to their population (Table 1), and (2) to quantify the confidence with which songs of individual males are assigned to a given population (Fig. 7). We interpret the former as a measure of overall divergence between two populations and the latter as a measure of song similarity of an individual to a population. As input, we used two recordings for each individual male (mean $\pm$ SD duration per recording: $6.8 \pm 1.6$ s, range 4.5–10.2 s; $n = 536$).

To quantify the overall classification success, we first trained the sound classifier on two categories of songs (e.g., songs of population $W_1$ versus $D_1$) using all available recordings from individuals from Generation 1 (i.e., 30-32 males per population, represented by 60–64 song recordings). After the training phase, the software reports a validation statistic, which is the proportion of training songs that are classified correctly with the algorithms derived from the training set (this value has to be interpreted cautiously, see below). For independent validation, we then tested the classification success (proportion of tested songs that are classified correctly) on recordings from individuals from Generation 2 (i.e., 17-20 males per population, using 34–40 songs). We did this separately for the males that had been cross-fostered within and between populations. All steps (training, validation, and testing) were carried out for all six pairwise combinations of the four captive populations used in this study.

Besides reporting a classification result for each tested recording, the sound classifier also reports a confidence statistic (complementary likelihoods of belonging to each of the two classes) for each 1 s interval of the recording in a sliding window with 50% overlap. As the classification success and overall confidence may increase with the length of recording, we trimmed all recordings to 4.5 s and averaged for each recording the confidence scores for a given class from the first (0 to 1 s) to the last (3.5 to 4.5 s) time interval. We interpret this mean confidence value in belonging to a certain class as a measure of similarity to that class. In analogy to the similarity values from SAP (see above), we retrieved 'ML similarity values' from the perspective of each female from Generation 2 with regard to the males from her rearing aviary. Hence, we trained the sound classifier to distinguish the songs of the eight parental males (Generation 1) of a female's rearing aviary from those of the other population type which the female would later encounter (e.g., $W_1$ vs $D_1$, 16 parental recordings each). The classifier was then tested with each of the songs of the (usually 40) males that the female would later encounter, to obtain values of their song similarity to the parents in her rearing aviary ('ML song similarity to parents'). The similarity values from each of the two recordings of a male were averaged (repeatability: $r = 0.88$, $n = 584$ pairs of values from 146 males, each combined with four female rearing aviaries). Similarly, we trained the sound classifier using the respective peer males of Generation 2 (males with whom females grew up in their rearing aviary) in contrast to peers from the other population type, to obtain values of similarity of males to those peer members ('ML song similarity to peers', repeatability $r = 0.91$, $n = 584$).

To further validate the classification procedure, we ran a negative control by training on two sets of 25 songs (mean duration 16.4 s per recording) from a single population. Classification success was 49.5% in the testing phase, which is close to the 50% chance level. Note that validation after training indicated an 80% classification ability within the training set, indicating that the utility of a trained classifier should be judged by independent testing and not from the validation percentages. We recorded all birds in one of two identical sound-proof chambers (see above), which ensured that classification success during testing stemmed from properties of the recorded songs rather than from idiosyncratic background noises. For example, such background noises might differ when wild populations would be recorded in their respective natural habitats. To avoid this problem, both sound-proof chambers were used about equally often in each of the four populations within each of the two recorded generations. As the two generations were recorded about 1 year apart, confounding effects of background noise that would closely match the offspring to their population of foster parents (depending on whether they had been cross-fostered between or within populations) can be excluded. No pre-processing of sound files (e.g., noise-reduction) was carried out prior to analysis.

**Data analysis**. To investigate whether pair-bonding and heterosexual social associations depended on culture (population of rearing) or on genetic background (population of origin), we used two statistical approaches. First, for the data set of identified pairs, we tested whether the observed degree of mating assortment by

either population of rearing or by population of origin differed from expectations under random mating (50:50), using an exact binomial test. We tested each replicate separately for each of the three generations.

Second, for the data set on heterosexual interactions (also including individuals that were defined as unpaired, see above), we constructed a social network, where nodes represented individuals and edges represented pair-bonding interactions between individuals. We did this separately for each aviary and for each breeding experiment (Generations 1–3). We then quantified the extent to which social interactions were clustered by culture by calculating the assortativity coefficient for each social network[66]. The assortativity coefficient is a network version of the Pearson's correlation coefficient, where the value from −1 to 1 reflects the tendency for individuals with similar attributes (here: population of rearing) to be associated in the network ($r = 1$), randomly associated ($r = 0$), or disassociated ($r = -1$). We used permutation tests to assess whether the association by culture was significantly non-random[44]. To obtain a $p$-value, we randomly re-allocated the phenotype value (population of rearing) across the nodes in the network (10,000 times) and calculated the assortativity coefficient for each permuted network. The $p$-value then equals the proportion of assortativity coefficients that were larger than the observed coefficient.

For the 'social experiment generation 2', we derived a daily social network using the pair-wise distance data and compiled this into a dynamic network video across the 30 days to visualise the association pattern. We also calculated the corresponding assortativity coefficients by culture for each day. Further, we analysed these daily social networks across 30 days within and between sexes to reveal the temporal patterns of assortment by song or by population of origin (genetic background) of each sex. This was done to investigate the differences of social patterns between heterosexual relationships and same-sex relationships.

We tested whether the daily pair-wise distance (from the social experiment Generation 2) can be explained by cultural (song) similarity and by genetic (size) similarity between females and males that participated in this social experiment. We used generalised mixed-effect models[67] with distance of each male-female combination as the response variable and with female identity (151 levels), male identity (151 levels), the combination of male and female identity (pair ID: 5752 levels), and the combination of the male's and the female's rearing aviaries (64 levels) as random effects. As fixed effects of interest, we fitted several categorical predictors that distinguish different types of male-female combinations (for details, see Supplementary Table 5) and several continuous predictors (measures of body size and song similarity, see above) that reflect individual-specific traits in a male-female combination.

**Reporting summary**. Further information on research design is available in the Nature Research Reporting Summary linked to this article.

## Data availability
All data including raw data for figures and tables are available at https://osf.io/d59rf/files/

## Code availability
All codes are available at https://osf.io/d59rf/files/

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

## Acknowledgements

We thank Henryk Milewski and Tomas Cordoba for carrying out the machine-learning analysis, Klaus Pichler and Felix Hartl for technical support, Nikolas Heinecke and Michael Harbich for IT support, Mihai Valcu for help with statistics and setting up the virtual machine, Sonja Bauer, Edith Bodendörfer, Jane Didsbury, Annemarie Grötsch, Andrea Kortner, Frank Lehmann, Petra Neubauer, Katharina Piehler, Frances Weigel and Barbara Wörle for animal care, and Jochen Wolf and Henrik Brumm for comments on the manuscript. This study was funded by the Max Planck Society (to BK). DW was supported by the CAS pioneer hundred talents programme (E1516511). DRF and LMA received additional funding from the Deutsche Forschungsgemeinschaft (DFG) Centre of Excellence 2117 'Centre for the Advanced Study of Collective Behaviour' under Germany's Excellence Strategy – EXC2117 – 422037984. DRF received additional funding from a DFG grant (FA 1420/4-1 awarded to DRF) and the European Research Council (ERC awarded to DRF) under the European Union's Horizon 2020 research and innovation programme (grant agreement No. 850859) and an Eccellenza Professorship Grant of the Swiss National Science Foundation (Grant Number PCEFP3_187058 awarded to DRF).

## Author contributions

D.W. initiated the study; W.F. and B.K. designed the study; D.W. and W.F. designed the experiments with input from D.F. and B.K.; D.W. collected the data with input from K.M. and Y.P.; D.W. recorded and analysed song with input from S.M., W.F. and Y.P.; D.F., A.M.C., G.A.N. and J.K. helped build the tracking system; D.W., W.F. and D.F. analysed the data; D.W., W.F., D.F., B.K., L.M.A. and A.M.C. wrote the manuscript.

## Funding

## Competing interests

The authors declare no competing interests.
