## [Peer Review File · Nature Communications]

Machine learning reveals cryptic dialects that explain mate choice in a songbirdEditorial Note: This manuscript has been previously reviewed at another journal that is not operating a transparent peer review scheme. This document only contains reviewer comments and rebuttal letters for versions considered at Nature Communications.

Reviewers Comments:

Reviewer #1 (Remarks to the Author):

I was reviewer 1 in the first round of revisions. I think the manuscript is very much improved over the first submission and I am really satisfied with the authors' responses and changes due to my comments in the first round. Reading this version of the manuscript, I have a number of new questions/concerns. My most serious concern is the conclusion that songs are the only culturally inherited trait that might influence mating patterns and that the experiment designed to test this is still only correlative. I want to note that I don't necessarily believe that my alternative explanation (outlined below in point 2) is correct, but, if the authors make the claim that cryptic song dialects guide mate choice, it is necessary to either conduct a proper experiment or remain much more cautious about which socially inherited traits influence mating patterns.

Major comments:

1) Lack of a null hypothesis for "cryptic song dialects". There is no definition given before the discussion for what the authors consider to be a dialect, which I think is an important omission in the introductory paragraphs. The definition given in the discussion is: sharp geographic transitions in vocal parameters. A devil's advocate question: wouldn't a machine learning algorithm be very likely to find differences in songs among many types of groups sufficient to classify them (e.g., family groups, mildly isolated populations lacking what would be called dialects)? What is the baseline expectation if there are really no "dialects"? Is a lack of dialects equivalent to absolutely no ability to classify songs? Work is needed to put their findings here into perspective, otherwise identifying "cryptic" dialects is possibly a given. Of course, if one believes the second part of the study, then there must be some song differences, but I am not completely convinced by this (see next comment).

In line with this is a paragraph in the discussion (lines 266-272) suggesting machine-learning as an alternative to testing discrimination ability. I strongly suggest that this paragraph be re-written, since measuring discrimination ability can never be replaced, only complemented, by metrics of signal dissimilarity. In some cases, large acoustic differences are apparently of little relevance to receivers, while, in others, small differences are critical. Additionally, I also very strongly disagree with the statement that machine learning is an alternative to measuring signal properties themselves, since it depends very much of the question, and the weaknesses of machine-learning approaches are not highlighted. For example, suppose a researcher wants to test which aspects of the signal are salient by manipulating/design an intermediate signal. Such an approach would require knowledge of which properties of the signals differ, not simply that they do differ!

2) All of the support for the argument that population-level song differences are the key driver of mate choice is correlative, since only rearing environment, and not song itself, is manipulated. A key argument that the authors make is that the only likely socially inherited trait that could influence social proximity and pairing is song. Figure 3, for example, states that the effect of rearing environment is transmission of songs and song preferences. However, despite the claim that they perform a trans-generational experiment to confirm song as the driving factor, their "experiment" is still correlative, since song, again, is not manipulated in isolation from other inherited factors. The main argument of the authors is that non-song learned behavioral traits have an open-ended learning

period (with no support given for this assumption that I can see) and then claim that other culturally inherited vocalizations are unlikely to explain the patterns they observe. However, distance calls are learned from the father and, although the authors argue that they have little linkage to songs, an older study showed that they are often incorporated themselves into songs (Zann 1990 *Animal Behaviour* 40:5). A counter-argument to the authors' claim that song differences drive mating pattern differences is the following: (a) distance calls are learned by offspring from parents and differ across populations, (b) distance calls are incorporated into songs, (c) differences in the distance calls incorporated into songs drive measured song differences across populations, (d) song differences do not matter for females, but distance calls do, (e) the apparent association between songs and female preferences is due to a correlation between distance calls and song form. Did the authors make note of the incorporation of distance calls during their acoustic analyses? If so, what happens to the machine learning and SAP approaches if they are eliminated from the songs?

Abstract

Lines 43-45: I found this sentence difficult to follow as written. I think the authors mean: "may lead to within population variation that is similar in magnitude to across population variation" Is that correct?

Lines 53-54: This part is a bit unclear. Do the authors mean songs? If not, what are they referring to?

Main

Lines 68-75: I have difficulties with this section. I really don't follow the argument that acoustic "space" can be filled out. I can see this for gross acoustic measures, like frequency range, but how would this work for syllable form/song syntax/etc? What's the support for this argument? If some species really don't have dialects, aren't high rates of dispersal a more likely explanation?

Line 81: Is this true generally in the wild or only in captivity? Both studies cited are in captivity.

Line 84: Isn't this due to high rates of dispersal?

Lines 97-98: How much isolation is there among populations in nature?

Lines 126-128: Why didn't the authors mix the two recently wild populations (and two domesticated populations) together? Given the artificiality of the domesticated populations and the authors claims about each zebra finch population "filling" acoustic space, it seems that the wild populations would be a much more interesting, naturally relevant comparison.

Reviewer #2 (Remarks to the Author):

This study is based on an impressively large-scaled, transgenerational mate choice experiments using genetically and culturally differentiated populations of zebra finches. In my previous assessment of this manuscript I found this manuscript highly interesting but asked the authors to deal with a few major clarifications. Unfortunately, these issues have not been fully addressed and I find some sections of the manuscript problematic.

1. I asked for a clarification of the main novel contribution to the field and the authors.

The authors argue that "First, we show that machine-learning algorithms can readily distinguish the songs of our four captive populations (Table 1), despite the fact that all trait-based analyses of zebra finch songs led to the conclusion that song differentiation was biologically insignificant [12, see also 46-48]. Second,

we show that zebra finches are sensitive to these differences, although the opposite had been predicted [12] for this “best-studied model” of song learning and perception.”

Comment: These are over-statements. Showing differences among the song of captive populations reveals a pattern (i.e. differences in dialects are often found) but does not per se provide evidence for a biological significance of these patterns. While the authors nicely show that rearing environment (cultural experiences) have a big effect on mate choice later in life, the underlying mechanism have not been revealed. The whole introduction and abstract focuses on song – but there is no direct test of the effect of song on mate choice.

2. I asked about the relevance of the social-network analyses in this context and for a description of how courtship proceeds and social bonds build up and leads to pairing. The authors have added information about how pairing happens but this description does not describe any role of song. This worry me as the argued main take-home-message from the study is that song plays a major role.

3. Another source of confusion is that the logic of the paper is outlined in a hypothesis testing way but the analyses use a “data-driven approach” making the results difficult to interpret. Some parts of the method section are also difficult to follow (e.g. if one wanted to repeat the experiment). For example, pairing patterns are monitored in large aviaries and the mate choice of any particular bird is therefor constrained by the mate choice made by other individuals in the same aviary (as the options decline). How is this problem solved by the analyses? It is also unclear when song is important during this process. Also, is mate choice mutual or is it only the females who decide?

Reviewer #3 (Remarks to the Author):

Re-review of “Machine learning reveals cryptic dialects that guide mate choice in a songbird”

My main issues with the previous version of this paper were the lack of demonstration of the differences in song, and misleading or overblown wording. Taking the second sort first, the study has been reoriented in a way that renders irrelevant a good many of the aspects of my main wording problems, and simple adjustments in word choice or tone have softened over others. The removal of some speculation to the extended data was a smart move, as was the change in focus of the paper as a whole, reducing the emphasis on speciation in favor of highlighting the sorts of features that guide mate choice and their developmental status, and (to a secondary degree for them but for me just as interesting) distinguishing between the hypotheses of widespread stochastic variation in song among individuals obliterating population signal and cryptic but real variation between populations. Regarding the song differences, their new machine learning analysis establishes differences between the songs, which is really the point of my earlier critique so I am mostly satisfied. I am still disappointed that we don't know what they are, but our ignorance there does not affect the interpretation. Finally, they still use the term “dialect”, which over geographical space is only one of three ways to manifest variation—stochastic variation (which they rule out), and clinal variation. If you pick songs from two populations that are sufficiently distant from each other and find that birds (or machine learning algorithms) can tell them apart, this does not establish dialects, it establishes difference, which could just as well be the result of geographically intervening continuous variation as the stepwise pattern that we call dialects. Nevertheless, this is not a geographic variation study and so there would be no way to distinguish between these two patterns among their populations. And “cryptic dialects” does have a ring to it that is not easily replaced. So I wouldn't make a stink about that. It should be mentioned that “cryptic” is based not on their own data, as (if memory serves) they do not analyze the songs conventionally and then fail to find a difference— rather, the authors claim the differences are cryptic

based on the failure of past studies to find population differences in other datasets. So again a slippery word, but not troubling enough to insist on a change. But, just as a parting shot, these two terms are in the very title, which thus has an element of "shooting the moon", as it includes the phrase "cryptic dialects" when the paper hasn't really shown the songs to be either cryptic or dialects. But it is a very good study, and in my opinion shows what it does in fact intend to show.

Reviewer #4 (Remarks to the Author):

This is a very interesting study, and relates well to current topics in animal learning and culture. I have been asked to comment specifically on the machine learning aspect of the work, so I focus only on that.

It is good that the acoustic analysis is conducted separately in two very different software programs (SAP and "ML", where "ML" here refers to Apple's CreateML). The ML analysis appears to perform much better at predicting the behavioural results, which is plausible given the current state of the art in ML.

My main concern on reading this paper was whether the acoustic data include confounding factors that might have been used by the black-box classifier. This is a common risk especially when the researchers do not exert control over which acoustic features are supplied to the classifier---this is certainly the case here, since we are not even able to find out which acoustic features are used by Apple CreateML, let alone to choose them. The controlled acoustic environment for sound recording reduces my concerns, assuming that the authors can confirm that the choice of recording chamber was not correlated with the class label. However, given that one of the main claims of this paper is that machine learning can discover cryptic attributes of song that guide mate choice, we should try to make sure that this claim is not spuriously generated by black-box ML's ability to latch on to confounding factors. The strong performance of ML versus SAP illustrated in Figure 2 is exactly the kind of outcome that could in some cases be an artefact of confounds.

Aside from this main issue, it appears to me that the ML part of the study has been conducted well.

My recommendation is thus: the paper is broadly very good, but that it cannot yet be published without taking a little more care to validate that the ML performance is genuine, and/or to improve the replicability of the ML process.

Some steps that can/should be taken in this regard:

Recording in identical soundproofed chambers (line 626) is indeed a very good way to eliminate many confounds. Could the authors confirm that the choice of which sound chamber to place a bird in was not correlated with the class label? The authors have argued that they are identical, but there may always be some small differences, e.g. in the sound of the ventilation fan or microphone placement.

Was any preprocessing applied to the audio data, e.g. noise-reduction to remove any machinery hum? This is not necessarily a good thing to do, but I would like to confirm.

I recommend optionally that the authors could validate their method is not being misled by background sound. For example using "data augmentation", adding low levels of white noise to the training and testing data and rerunning the automatic classification. (Stowell/Petrusková/Šálek/Linhart 2019 focus on these questions, applying a fairly complicated version of data augmentation.) This is an optional step, but would be compulsory if the choice of sound chamber is in fact correlated with the

class label.

It is not clear what are the acoustic features or method used by Apple CreateML. This reduces replicability. At the very least the authors must document the software version of CreateML used. Ideally the authors should state something more about what type of machine learning the framework uses. Apple's documentation does not give much information, which reduces the value of that tool for scientific use.

Data availability -- I strongly recommend to publish the sound recordings, generally for open science, but more specifically so that other more interpretable machine-learning models can be used to find out what aspects of the data lead to this song discriminability. ("Explainable" algorithms are a current topic in deep learning, and could eventually help to tell us what are the discriminating features here.)

A further option would be to apply an additional ML algorithm, one that is open-source and well-documented in scientific literature, to replicate the findings obtained using CreateML. I do not insist on this. However, it is one obvious way to fix the opacity of method.

Out of the above options, my core recommendation is to publish the acoustic data openly, to validate the classifier further using data augmentation with low-level white noise, and to document the CreateML version along with any further info they can find.

Minor comments:

line 269 "closer to the biological reality of training a neural network" - What is "the biological reality of training a neural network"? This phrase seems rhetorical and needs unpacking.

line 543 "principle component" -> "principal component"

Reviewers Comments:

Reviewer #1 (Remarks to the Author):

I was reviewer 1 in the first round of revisions. I think the manuscript is very much improved over the first submission and I am really satisfied with the authors' responses and changes due to my comments in the first round. Reading this version of the manuscript, I have a number of new questions/concerns. My most serious concern is the conclusion that songs are the only culturally inherited trait that might influence mating patterns and that the experiment designed to test this is still only correlative. I want to note that I don't necessarily believe that my alternative explanation (outlined below in point 2) is correct, but, if the authors make the claim that cryptic song dialects guide mate choice, it is necessary to either conduct a proper experiment or remain much more cautious about which socially inherited traits influence mating patterns.

Reply: In principle, we agree with the reviewer that an experiment in which one manipulates only the song of the birds would be ideal for obtaining strong evidence (see line 237 in the tracked version). However, such an experiment is not possible, because one cannot achieve full experimental control over what males sing, and the use of robotic male finches in experiments on female mate choice is not sufficiently naturalistic to result in pair formation. Hence, cross-fostering – the method we used – remains the only feasible experimental manipulation of song. When using cross-fostering, it is essential to rule out confounding factors that might be correlated with the manipulation of song. The evidence for song being causal is correlational, but about as strong as it can possibly be, with Table 2 providing two independent approaches yielding the same result (see line 229-240 & 261-262). We can also rule out the reviewer's alternative scenario that distance calls or courtship dance would be the true target of female choice and lead to spurious correlations with song (see lines 262-272). In light of these arguments, which we added to the manuscript, we feel that the writing is sufficiently cautious, pointing out that the evidence is correlational rather than strictly experimental (see lines 233-236).

Major comments:

1) Lack of a null hypothesis for "cryptic song dialects". There is no definition given before the discussion for what the authors consider to be a dialect, which I think is an important omission in the introductory paragraphs. The definition given in the discussion is: sharp geographic transitions in vocal parameters. A devil's advocate question: wouldn't a machine learning algorithm be very likely to find differences in songs among many types of groups sufficient to classify them (e.g., family groups, mildly isolated populations lacking what would be called dialects)? What is the baseline expectation if there are really no "dialects"? Is a lack of dialects equivalent to absolutely no ability to classify songs? Work is needed to put their findings here into perspective, otherwise identifying "cryptic" dialects is possibly a given. Of course, if one believes the second part of the study, then there must be some song differences, but I am not completely convinced by this (see next comment).

Reply: We appreciate the comment of the reviewer, but in our view, a discussion about definitions of what constitutes a 'dialect' is not particularly useful. The definition of 'dialect' has been debated previously, and often researchers have propagated a definition that fitted their ability to measure something. With the use of 'machine learning' approaches, such earlier methodological definitions seem outdated, and we feel that proposing a new definition is not helpful. We would instead prefer to emphasise the striking observation that song differences (that have previously

been declared as likely irrelevant [12]) appear to be highly salient to the birds. If song properties of smaller groups like families would turn out to have similarly striking effects on behaviour, we do not see why one could not for instance speak of ‘family dialects’. Hence, we find the term ‘cryptic dialects’ helpful to explain the (rather unexpected) findings, and we do not feel that we are thereby violating an important definition.

In line with this is a paragraph in the discussion (lines 266-272) suggesting machine-learning as an alternative to testing discrimination ability. I strongly suggest that this paragraph be re-written, since measuring discrimination ability can never be replaced, only complemented, by metrics of signal dissimilarity. In some cases, large acoustic differences are apparently of little relevance to receivers, while, in others, small differences are critical. Additionally, I also very strongly disagree with the statement that machine learning is an alternative to measuring signal properties themselves, since it depends very much of the question, and the weaknesses of machine-learning approaches are not highlighted. For example, suppose a researcher wants to test which aspects of the signal are salient by manipulating/design an intermediate signal. Such an approach would require knowledge of which properties of the signals differ, not simply that they do differ!

Reply: We fully agree with the reviewer and have re-written this paragraph accordingly (lines 304-320). We now emphasise that behavioural assays cannot be replaced by metrics of signal dissimilarity (lines 308-309). We also added the reviewer’s comment about machine learning having the disadvantage of not yielding diagnostic song parameters (lines 318-320).

2) All of the support for the argument that population-level song differences are the key driver of mate choice is correlative, since only rearing environment, and not song itself, is manipulated. A key argument that the authors make is that the only likely socially inherited trait that could influence social proximity and pairing is song. Figure 3, for example, states that the effect of rearing environment is transmission of songs and song preferences. However, despite the claim that they perform a trans-generational experiment to confirm song as the driving factor, their “experiment” is still correlative, since song, again, is not manipulated in isolation from other inherited factors. The main argument of the authors is that non-song learned behavioral traits have an open-ended learning period (with no support given for this assumption that I can see) and then claim that other culturally inherited vocalizations are unlikely to explain the patterns they observe. However, distance calls are learned from the father and, although the authors argue that they have little linkage to songs, an older study showed that they are often incorporated themselves into songs (Zann 1990 Animal Behaviour 40:5). A counter-argument to the authors’ claim that song differences drive mating pattern differences is the following: (a) distance calls are learned by offspring from parents and differ across populations, (b) distance calls are incorporated into songs, (c) differences in the distance calls incorporated into songs drive measured song differences across populations, (d) song differences do not matter for females, but distance calls do, (e) the apparent association between songs and female preferences is due to a correlation between distance calls and song form. Did the authors make note of the incorporation of distance calls during their acoustic analyses? If so, what happens to the machine learning and SAP approaches if they are eliminated from the songs?

Reply: As explained above, full experimental control over male song cannot be achieved. Our cross-fostering experiment achieves manipulation, but not full control and not without possible confounds. Hence, we agree that it is essential to discuss such possible confounds. To address the specific scenario of the distance call being such a possible confound, we have added details about the extent to which a male’s distance call resembles his song. In a previous study, we analysed distance calls and song of 413 males from one of our four study populations, and we found only a

weak correspondence between a male's call and his song (mean Pearson correlation $r=0.16$, range 0.04 – 0.24, $n=16$ parameters, including 8 characteristics of voice; Forstmeier et al. 2009). Zann (1990) found that distance calls were incorporated into male's songs, but this is not the own distance call, but those of other individuals. With such weak correlations between call and song of the same individual, the findings on the effect of songs reported in Table 2 cannot emerge as spurious correlates of the male's call characteristics and the latter thus cannot be the true target of female choice. We now explain in more detail why it is unlikely that our findings on songs are confounded by effects of calls or of courtship dance (in lines 262-272).

Abstract

Lines 43-45: I found this sentence difficult to follow as written. I think the authors mean: "may lead to within population variation that is similar in magnitude to across population variation" Is that correct?

Reply: Correct. We have rephrased this sentence to make it clear (lines 44-46).

Lines 53-54: This part is a bit unclear. Do the authors mean songs? If not, what are they referring to?

Reply: We have removed this sentence, as it was not necessary (line 54).

Main

Lines 68-75: I have difficulties with this section. I really don't follow the argument that acoustic "space" can be filled out. I can see this for gross acoustic measures, like frequency range, but how would this work for syllable form/song syntax/etc? What's the support for this argument? If some species really don't have dialects, aren't high rates of dispersal a more likely explanation?

Reply: The idea is explained in more detail in the two references that we cite [12, 13]. To clarify that this is not necessarily a matter of dispersal, we have added the explanation that this may come about through high rates of song innovation (lines 84-85). Dispersal may be an additional factor in the wild, but this would not apply to the captive populations that these studies refer to.

Line 81: Is this true generally in the wild or only in captivity? Both studies cited are in captivity.

Reply: This is also true for the wild, which we now explicitly state and show by adding a citation of a study from the wild (line 94).

Line 84: Isn't this due to high rates of dispersal?

Reply: Yes, this is possible, but here we are not aiming at explaining why.

Lines 97-98: How much isolation is there among populations in nature?

Reply: This is not so clear. Two studies in the wild have found some geographic differentiation in songs and in distance calls (Zann 1993, Runciman et al. 2005).

Lines 126-128: Why didn't the authors mix the two recently wild populations (and two domesticated populations) together? Given the artificiality of the domesticated populations and the authors claims about each zebra finch population "filling" acoustic space, it seems that the wild populations would be a much more interesting, naturally relevant comparison.

Reply: The isolation between the two recently wild-derived populations is just shorter (and hence likely to yield smaller effects), but it is equally artificial as it also happens in captivity.

Reviewer #2 (Remarks to the Author):

This study is based on an impressively large-scaled, transgenerational mate choice experiments using genetically and culturally differentiated populations of zebra finches. In my previous assessment of this manuscript I found this manuscript highly interesting but asked the authors to deal with a few major clarifications. Unfortunately, these issues have not been fully addressed and I find some sections of the manuscript problematic.

1. I asked for a clarification of the main novel contribution to the field and the authors.

The authors argue that “First, we show that machine-learning algorithms can readily distinguish the songs of our four captive populations (Table 1), despite the fact that all trait-based analyses of zebra finch songs led to the conclusion that song differentiation was biologically insignificant [12, see also 46-48]. Second, we show that zebra finches are sensitive to these differences, although the opposite had been predicted [12] for this “best-studied model” of song learning and perception.”

Comment: These are over-statements. Showing differences among the song of captive populations reveals a pattern (i.e. differences in dialects are often found) but does not per se provide evidence for a biological significance of these patterns. While the authors nicely show that rearing environment (cultural experiences) have a big effect on mate choice later in life, the underlying mechanism have not been revealed. The whole introduction and abstract focuses on song – but there is no direct test of the effect of song on mate choice.

Reply: We respectfully disagree with this view. Our study is novel because it finds that the songs of the studied populations are clearly distinguishable (Table 1), which stands in clear contrast to claims of earlier studies stating that there is little differentiation and that these differences would unlikely matter to the birds (see [12]; see lines 84-92). Moreover, Table 2 shows strong effects of song similarity on mate choice.

2. I asked about the relevance of the social-network analyses in this context and for a description of how courtship proceeds and social bonds build up and leads to pairing. The authors have added information about how pairing happens but this description does not describe any role of song. This worry me as the argued main take-home-message from the study is that song plays a major role.

Reply: We find the additional social network analyses (Figure 5) insightful with regard to the temporal pattern and the sex-specificity, as this rules out some alternative explanations that are not based on song. Direct effects of song on pairwise-distances are shown in Table 2.

3. Another source of confusion is that the logic of the paper is outlined in a hypothesis testing way but the analyses use a “data-driven approach” making the results difficult to interpret.

Reply: We have separated Figure 3, which shows our *a priori* hypotheses, from the Supplementary Figure 7, which illustrates the data-driven approach.

Some parts of the method section are also difficult to follow (e.g. if one wanted to repeat the experiment). For example, pairing patterns are monitored in large aviaries and the mate choice of any particular bird is therefor constrained by the mate choice made by other individuals in the same aviary (as the options decline). How is this problem solved by the analyses? It is also unclear when song is important during this process. Also, is mate choice mutual or is it only the females who decide?

Reply: We agree that these are interesting aspects, and we would welcome constructive suggestions of where our manuscript could be improved.

Reviewer #3 (Remarks to the Author):

Re-review of “Machine learning reveals cryptic dialects that guide mate choice in a songbird”

My main issues with the previous version of this paper were the lack of demonstration of the differences in song, and misleading or overblown wording. Taking the second sort first, the study has been reoriented in a way that renders irrelevant a good many of the aspects of my main wording problems, and simple adjustments in word choice or tone have softened over others. The removal of some speculation to the extended data was a smart move, as was the change in focus of the paper as a whole, reducing the emphasis on speciation in favor of highlighting the sorts of features that guide mate choice and their developmental status, and (to a secondary degree for them but for me just as interesting) distinguishing between the hypotheses of widespread stochastic variation in song among individuals obliterating population signal and cryptic but real variation between populations. Regarding the song differences, their new machine learning analysis establishes differences between the songs,

which is really the point of my earlier critique so I am mostly satisfied. I am still disappointed that we don't know what they are, but our ignorance there does not affect the interpretation. Finally, they still use the term “dialect”, which over geographical space is only one of three ways to manifest variation— stochastic variation (which they rule out), and clinal variation. If you pick songs from two populations that are sufficiently distant from each other and find that birds (or machine learning algorithms) can tell them apart, this does not establish dialects, it establishes difference, which could just as well be the result of geographically intervening continuous variation as the stepwise pattern that we call dialects. Nevertheless, this is not a geographic variation study and so there would be no way to distinguish between these two patterns among their populations. And “cryptic dialects” does have a ring to it that is not easily replaced. So I wouldn't make a stink about that. It should be mentioned that “cryptic” is based not on their own data, as (if memory serves) they do not analyze the songs conventionally and then fail to find a difference— rather, the authors claim the differences are cryptic based on the failure of past studies to find population differences in other datasets. So again a slippery word, but not troubling enough to insist on a change. But, just as a parting shot, these two terms are in the very title, which thus has an element of “shooting the moon”, as it includes the phrase “cryptic dialects” when the paper hasn't really shown the songs to be either cryptic or dialects. But it is a very good study, and in my opinion shows what it does in fact intend to show.

Reply: We thank the reviewer for these interesting thoughts. After considering the pros and cons we would prefer to keep the “cryptic dialects” in the title of our manuscript, as this likely will attract the interest of potential readers (see also our reply to a comment from Reviewer 1 above).

Reviewer #4 (Remarks to the Author):

This is a very interesting study, and relates well to current topics in animal learning and culture. I have been asked to comment specifically on the machine learning aspect of the work, so I focus only on that.

It is good that the acoustic analysis is conducted separately in two very different software programs (SAP and "ML", where "ML" here refers to Apple's CreateML). The ML analysis appears to perform much better at predicting the behavioural results, which is plausible given the current state of the art in ML.

My main concern on reading this paper was whether the acoustic data include confounding factors that might have been used by the black-box classifier. This is a common risk especially when the researchers do not exert control over which acoustic features are supplied to the classifier---this is certainly the case here, since we are not even able to find out which acoustic features are used by Apple CreateML, let alone to choose them. The controlled acoustic environment for sound recording reduces my concerns, assuming that the authors can confirm that the choice of recording chamber was not correlated with the class label. However, given that one of the main claims of this paper is that machine learning can discover cryptic attributes of song that guide mate choice, we should try to make sure that this claim is not spuriously generated by black-box ML's ability to latch on to confounding factors. The strong performance of ML versus SAP illustrated in Figure 2 is exactly the kind of outcome that could in some cases be an artefact of confounds.

Reply: We thank the reviewer for his insightful comments on ML. We are aware of the problem that confounding background noise could lead to a spurious assignment of songs to the correct population (lines 677-681). We therefore had allocated the birds in about equal numbers to the two recording chambers (as we now explicitly state in lines 681-683). Moreover, it should be noted that the birds of generation 2 were recorded about one year after the birds of generation 1. Thus, it seems highly unlikely that any confounding background noise could have led to a nearly perfect assignment of each rearing cohort (16 natal aviaries) to their correct foster environment (4 dialects). This is now also explained in lines 683-686.

Aside from this main issue, it appears to me that the ML part of the study has been conducted well.

My recommendation is thus: the paper is broadly very good, but that it cannot yet be published without taking a little more care to validate that the ML performance is genuine, and/or to improve the replicability of the ML process.

Some steps that can/should be taken in this regard:

Recording in identical soundproofed chambers (line 626) is indeed a very good way to eliminate many confounds. Could the authors confirm that the choice of which sound chamber to place a bird in was not correlated with the class label? The authors have argued that they are identical, but there may always be some small differences, e.g. in the sound of the ventilation fan or microphone placement.

Reply: Yes, this is now confirmed in lines 681-683.

Was any preprocessing applied to the audio data, e.g. noise-reduction to remove any machinery hum? This is not necessarily a good thing to do, but I would like to confirm.

Reply: There was no pre-processing of the recordings, as we now explicitly mention in lines 686-687.

I recommend optionally that the authors could validate their method is not being misled by background sound. For example using "data augmentation", adding low levels of white noise to the training and testing data and rerunning the automatic classification.

(Stowell/Petrusková/Šálek/Linhart 2019 focus on these questions, applying a fairly complicated version of data augmentation.) This is an optional step, but would be compulsory if the choice of sound chamber is in fact correlated with the class label.

Reply: Given that we see no risk of confounding background noise (see above), we consider this step unnecessary.

It is not clear what are the acoustic features or method used by Apple CreateML. This reduces replicability. At the very least the authors must document the software version of CreateML used. Ideally the authors should state something more about what type of machine learning the framework uses. Apple's documentation does not give much information, which reduces the value of that tool for scientific use.

Reply: We have added more information on the version and working environment in lines 630-631.

Data availability -- I strongly recommend to publish the sound recordings, generally for open science, but more specifically so that other more interpretable machine-learning models can be used to find out what aspects of the data lead to this song discriminability. ("Explainable" algorithms are a current topic in deep learning, and could eventually help to tell us what are the discriminating features here.)

Reply: We fully agree and will make all sound recordings freely available on the Open Science Framework (OSF) platform. This should allow for additional insights when new tools become available.

A further option would be to apply an additional ML algorithm, one that is open-source and well-documented in scientific literature, to replicate the findings obtained using CreateML. I do not insist on this. However, it is one obvious way to fix the opacity of method.

Out of the above options, my core recommendation is to publish the acoustic data openly, to validate the classifier further using data augmentation with low-level white noise, and to document the CreateML version along with any further info they can find.

Reply: As our manuscript is already somewhat complex due to the use of two methods of sound analysis (see Figure 6), we would prefer not to add a third method that only serves to replicate the results from one of the two methods. With the sound files being made freely available, future re-analysis with different existing and still-to-be-developed tools will be possible.

Minor comments:

line 269 "closer to the biological reality of training a neural network" - What is "the biological reality of training a neural network"? This phrase seems rhetorical and needs unpacking.

Reply: We modified this statement accordingly (lines 315-316).

line 543 "principle component" -> "principal component"

Reply: Thanks for spotting this error. Fixed (line 589).

Reviewers' Comments:

Reviewer #1:

Remarks to the Author:

In this study, the authors conducted a transgenerational experiment to test whether subtle differences in song between captive populations could persist across generations, and whether these differences also affect mating behaviour. The authors used four populations (two domesticated and two wild-derived) and ran a cross fostering experiment to disentangle culturally inherited traits from genetically inherited traits. The authors then used a novel machine learning methodology to show that there were subtle but significant differences between the songs of the populations, and that these differences persisted across generations. Finally, the authors showed that the song differences were associated with affiliative behaviour of females, such that females preferred to associate with males that sang songs that were similar to songs of the males from the rearing colony of the females.

This paper describes an interesting and novel study, and it is well written. There are, however, some issues that need to be addressed before the paper is ready to be published. Provided that these changes are made, I look forward to seeing this paper published in the future. I have explained my concerns below, but I also agree with some of the issues that were previously raised by the other reviewers.

Comments for the authors

Even though your results do not seem to be explained by song alone, you only briefly mention that you designed two post-hoc hypotheses to better explain the observed data. These hypotheses are interesting, but you do not discuss them sufficiently. You show a strong effect of rearing environment (which you assume to reflect only song learning) but that there is also an effect of phenotype. My point is that this could be due to the size of the birds, but it could also be other physiological traits that you did not measure. For example, there are sometimes clear deviations from wild-type coloration in domesticated zebra finches, and domesticated male zebra finches have higher testosterone levels compared to their wild counterparts (see Prior et al 2017 *Gen Comp Endocrinol*). I am not suggesting that you should measure these traits, but I would like to see some discussion of what other physiological traits may explain your results. This is particularly important considering that body size did not appear to explain your data very well (eg lines 211-212). On line 128 you mention that there may be other morphological traits, but this is not enough of a discussion. You also discuss your post-hoc hypotheses in the supplementary materials, but the manuscript would benefit from some of this discussion being incorporated in the main text.

There are potentially many other behavioural traits that zebra finches inherit from their rearing parent, and some of these traits could affect social dynamics. The manuscript would benefit from a more detailed discussion of these traits. For example, song rate is affected by the developmental environment in zebra finches (Tschirren 2009 *J Evol Biol*). One of your reasons for not discussing these possibilities (judging by your reply to previous comments) is that you can't achieve complete experimental control of song in this experiment. However, you could relatively easily test female preference experimentally through a playback experiment. Directly testing female song preference may be beyond the scope of this study, but you still need to discuss the limitations of your paper and alternative mechanisms that could explain your results.

I agree with previous reviewers that the use of "cryptic dialects" is problematic. You have not defined this term, and I do not understand the reason why you use it. "Subtle population differences in song" would be easier for a reader to intuitively understand. If you can find a good scientific justification for

the term (and explain it in the manuscript), I have no issue with it being used.

Minor comments

I had not heard of zebra finches not having dialects because of their variable song. This is an interesting hypothesis, but I am not entirely convinced. The zebra finch does not have a song that is particularly complex compared to species with larger repertoires. Also, some signs of dialects have already been described in this species (you cite this research). I suggest changing how you discuss the novelty of your results.

You kept these birds in groups, meaning that the birds could learn open-ended and close-ended behavioural traits from several individuals. Zebra finches preferentially learn song from their social father, but they also learn song from other members of the colony. Did you gather any data on which potential tutors the cross fostered males associated with?

How did you determine mate choice in the breeding experiment? Was it based on pair-bonding behaviour, barcode tracking, or both? Since you housed the birds in aviaries and you could observe them, it should have been easy to look at pair-bonding behaviour as well as mating behaviour (nest building, provisioning, etc). For example, how many of the 60 pairs from the first breeding experiment produced (genetic or EPC) offspring? If you are drawing conclusions about mate choice, this information would be highly relevant to this paper.

Line 76 "acoustic space" is not a particularly intuitive term. This sentence would be easier to understand if it read something like: "In such cases, the need for individual recognition and distinctiveness may favour individuals that produce innovative or unusual songs, thereby eliminating the potential for local dialects."

81-84 I do not agree that this is the "prevailing view". By doing a rather quick search, I was able to find at least one other study that used zebra finches to create salient (albeit artificial) "dialects" in captive populations. (Le Maguer 2021 Ethology). The results from Zann (1993) suggests that there are some differences in song between wild populations. These differences may be slight, but they were shown using traditional methods. It is interesting that you cite Lachlan here, considering that this paper did find differences between population, but they did not want to draw conclusions about the signal's salience. It would be more accurate to say that only subtle population have been found in this species, and that it is unclear whether these differences are salient. I suggest that you cite Le Maguer here (and where it suits).

128 If you are mentioning "unmeasured aspects of phenotype" you should also mention that there are possible culturally inherited traits that you have not measured.

191-196 I do not understand why the best explanation for the observed data has been moved to the supplementary materials. You should include this in the main text.

224-225 You do not have to achieve full experimental control over songs to experimentally test female preference. For example, a playback experiment could determine if the females can discriminate between these dialects.

Line 228-241. If they learn other aspects of behaviour than just song (such as foraging patterns or song rate), these traits could be passed on to the subsequent generation just as much as song. I would recommend using this paragraph to discuss your post hoc hypotheses. It is interesting that you found an effect of domesticated phenotype in addition to song, and this is something that should be discussed in the main text. Alternatively, you could use this paragraph to discuss what other traits

may be culturally inherited in this species, and the underlying mechanisms.

259 I disagree with this statement. Your study shows that culturally inherited differences between populations affect assortative mating behaviour in zebra finches. You do not directly test if song is the mechanism for this assortative mating.

263 Again, it seems to me that most papers on this topic use a much more cautious language when discussing dialects in zebra finches. When you use phrases such as “the entire space of acoustic and syntactical possibilities” that indicates that there are no dialects, but the papers you cite have found small differences between populations.

Line 266: Differences in song can be either cryptic or striking, they cannot be both. Please change the wording here, and carefully look over the manuscript to make sure the wording is correct and consistent.

268 Add a citation to this statement.

271 Do you mean that zebra finches belong to this group? I would argue that many songbirds have a much more complex song than the zebra finch. Zebra finches may have a variable song (large differences between individuals), but that is not the same as having a complex song (long and variable song motifs or a large repertoire).

276-279 Please find a better motivation for the use of “cryptic dialects”. Previous studies have found subtle song differences between populations, and your machine learning approach also finds subtle (but salient) differences between populations. What justifies “cryptic dialect” in this case? Do dialects have to be salient to females according to your definition?

Line 291: I do not agree that it is arguably closer to the biological reality. Often machine learning approaches are sensitive to small differences, but they are also a black box. In order to claim that a method is “close to biological reality” it would need to take into account the sensory biology of the animal model. A machine learning approach would easily find differences that cannot be perceived by the animal and there would be no way for the researcher to know this without using a behavioral approach.

598-604 I have not heard of using SAP like this. My guess is that you only looked at overall similarity? How does these “super motifs” compare to a single song in terms of sequential similarity, accuracy, etc.? Did you work directly with the developers of SAP when designing

Reviewer #2:

Remarks to the Author:

I was Reviewer 1 during the review process for Nature Ecology and Evolution. The aim of the study is to explore whether the songs of four captive populations of zebra finches have subtle, but salient differences to females. The authors conclude that there are subtle song differences among populations (which has been long known, as stated by the authors in lines 86-89) and that these differences guide female mate choice (which is the main novel claim of the paper). Carrying over from my previous review, I have issues with both of these conclusions. This was a hard review to write, because I think the authors' findings are really interesting. However, their conclusions go way beyond the data they actually have. Without an experiment isolating the influence of other environmental factors, I don't think that the authors can claim song differences determine mate choice.

1) I was surprised and somewhat troubled to read the authors' statements that it is not necessary to define what a dialect is and that defining a dialect is now "outdated" given machine learning approaches. There are two issues here.

First, there is a serious intellectual issue: the authors are bold enough to use the term "cryptic song dialect" to, as they write "attract the interest of potential readers", but not willing to actually state what they mean by the terms "dialect" or "cryptic".

Second, the authors themselves seem confused about what a dialect is: On the one hand, they claim that machine learning is sufficient to prove dialects, which seems to be implicitly arguing that any detectable difference between any two groups constitutes a dialect. In this case, as I wrote in the previous round of reviews, it is likely that machine learning approaches would demonstrate differences on many types of arbitrary groupings of individuals. What is the null hypothesis for cryptic dialects? Without this, it is impossible in my view to conclude whether the authors' results are at all surprising. (And they are indeed not, since earlier work has already shown some geographic differences in zebra finch songs). On the other hand, the authors say that any song variation associated with response variation constitutes a dialect. But we know that song variation within a population exists and has behavioral effects—are we now supposed to say that there are within-population and within-family group dialects as long as the songs from one group are preferred over those from another?

2) I was also surprised to read the authors' statement that an experiment to test whether song preferences guide mate choice is not possible. What about testing female discrimination and/or preferences for songs? This would clearly be able to isolate song as a factor influencing mate choice. I realize that such experiments would obviously be some work to complete (and maybe not possible given the state of the colonies), but there are clearly options aside from constructing robot birds!

3) Although the authors feel that they have toned down their conclusions, statements such as "Our study shows that population-specific song dialects drive strong assortative mating in zebra finches" and "we reveal experimentally that these 'cryptic song dialects' have real consequences for social behaviour" and "we show that zebra finches are surprisingly sensitive to population differences in song during the process of mate choice" are not at all cautious. The manuscript repeatedly equates culture with song e.g., "Hypothesis 3 assumes a learnt preference for a cultural trait, such that all birds prefer to mate with a partner from their own cultural population because of socially transmitted variation in song characteristics." Without a proper experiment showing, making this conclusion depends on eliminating any reasonable alternative explanation. However, I am unconvinced by the authors' arguments that alternative learned or inherited behavioral traits could not play a critical role in mate choice. The authors' main argument against this is that they know that song has a fixed sensitive period and then they assume that any other behavioral trait has an open-ended learning period (line 232) and would be mostly learned from the social group rather than the parents. Aside from the lack of any evidence that other learned traits are open-ended, this argument could not eliminate other inherited effects that might be correlated with song, such as early stress.

However, given that I brought it up in a previous review, I'll specifically discuss distance calls being incorporated into the song. Distance calls are included in the song (in fact often the loudest element), are learned from the parents, and expresses variation both geographically as well as between lab and wild populations. Moreover, the acoustic structure of the distance call strongly influences social-behavioral responses (Vignal and Mathevon 2011 in *J Comp Psych*). Why couldn't then the inherited distance call, which is also found in the song, play a key role in social interactions and mate choice?

The authors' main argument here is an earlier study showing a weak acoustic correlation between a

male's own distance call and his song. There are a few issues with this argument. First, this doesn't at all address the question of whether the distance call in the song is itself similar to the distance calls heard early in life, which is the relevant comparison. Second, I don't see how the correlation between a signal element and the whole song is appropriate. Certain elements within or associated with songs may play a key role in responses, rather than the overall acoustic characteristics of the song. For example, the whistle in white-crowned sparrow songs or the chatter call in cowbird songs. The whistle alone might have a low acoustic correlation with an entire white-crowned sparrow song, but this does not mean the whistle is not found in the song. The relevant comparison here would be to identify songs having the distance call as introductory or concluding element and compare that with the distance calls of the early social environment.

I think that the authors can conclude no more than that: 1) there are some differences in the songs of the groups they've studied and 2) there is some trait that is influenced by early life that is itself correlated with song (and might possibly be the song or some associated vocal trait) that influences mate choice.

Minor comments:

Abstract

Lines 55-56: Isn't this extremely well-known? Essentially the authors are saying that human observers haven't previously detected differences that are relevant to the birds themselves, but I think this is widely accepted.

Introduction

Overall, the introduction is very zebra finch focused—is this really a technical zebra finch-researcher question and study or of broader relevance? I think the authors could do more to broaden the appeal of this study.

Lines 106-110: why is sexual imprinting contrasted with song learning? Can't song learning be understood as a kind of sexual imprinting?

Line 110: why the term "cultural" drift? Couldn't there be environmental reasons? Couldn't there be variation in receivers that make certain song types more effective?

Discussion

Lines 256-257: This is incredibly jargony. I honestly have no idea what is meant by "each zebra finch population covers the entire space of acoustic and syntactical possibilities". Do the authors mean that every population, no matter how small or uniform produces all possible songs???

Line 260: throughout, the words "subtle" and "cryptic" are used, but now the differences are referred to as "striking"—based on what?

Line 262: is this a case of inadequate sampling or a true fact?

Line 283: "closer to the biological reality of an individual brain that distinguishes familiar from unfamiliar" What does this mean? And how does machine learning do this better than actually testing what a female responds to? Wouldn't a discriminant analysis on a suite of song traits do the same?

Line 287: I would add that it might also pick up non-relevant differences in the songs due to background noise, recording techniques, etc... etc...

Reviewer #3:

Remarks to the Author:

Overall, the authors appear to have taken good coherent actions to respond to the reviewers.

In this round, I again concentrate primarily on the machine learning aspect.

The authors have correctly addressed one of my main concerns (the possibility of confounding acoustic factors). I now have no objection about that.

My other main concern is the opacity of the ML used (Apple's "CreateML") which creates a risk against reproducibility of the results, or the ability of future researchers to apply the same approach even when this software version is changed/gone. The researchers have provided software version information, which is helpful; and with the publication of the audio data (as pledged) and associated information, I am happy to accept the claim that adding another ML classifier would overcomplicate the paper. I remain concerned that this aspect is not conducive to open science, and I hope that in future the analysis can be performed using a fully open software approach.

Thus, I am content to recommend the paper be accepted.

Reviewer #4:

Remarks to the Author:

I have reviewed this manuscript before (as reviewer 2) and some main issues remain:

1. There is no direct test for the effect of song, per se, on mate choice. Table 2 reveals a correlation but cannot rule out effects of other socially inherited traits (there may be many other "cryptic" traits).
2. It is unclear when song is important during the pairing process. (Information lacking about the pairing process)
3. Pairing patterns are monitored in large aviaries meaning that the mate choice of any particular bird is contained by the mate choice made by other individuals. How is this problem solved by the analyses?

REPLY TO REVIEWER COMMENTS

General reply to the comments of Reviewers #1, #2, and #4

Three reviewers (#1, #2, and #4) comment on our study allegedly lacking “a proper experimental test” showing that song dialects are causal to the observed patterns of assortative mating. We argue that the analysis presented in Table 2 of the manuscript presents much stronger evidence for the claim that dialects drive assortative mating than the playback experiment suggested by the reviewers would. Specifically, we explain (1) why playback experiments cannot establish causality and (2) why the results from Table 2 represent strong evidence.

- (1) In our study, we show a strong relationship between rearing environment as a binary trait (the predictor variable X) and pairing patterns (the dependent variable Y). The question at hand is whether this relationship is caused by effects of song dialects (X1) or other culturally inherited traits (X2). Assume we would conduct a playback experiment, in which we would demonstrate that playback of dialects (X1) has an effect on a variable such as the time spent close to a loudspeaker (a dependent variable Z). Such a finding would only be convincing evidence for the hypothesis that dialects drive assortative mating *if* variables Y and Z are equivalent (i.e. strongly correlated). However, this clearly remains a matter of interpretation and there are good reasons to doubt that the outcome of such playback experiments is indicative of mating preferences.

Let us consider the three main playback setups used in previous studies in more detail to explain the assertion that playback experiments do not (necessarily) indicate mating preferences. (A) In a phonotaxis test, a female is isolated from the group and placed between two speakers, either of which she can approach. Choice is then defined as the proportion of time spent near each speaker. In such tests, females typically spend more time close to the more familiar sounds (mate 70-95% (N. Clayton, 1988; Miller, 1979a; Woolley & Doupe, 2008), father 65-70% (N. Clayton, 1988; Miller, 1979b), normal rather than abnormal song 69% (Svec & Wade, 2009), and conspecific rather than heterospecific song 60-75% (D. Campbell & M. Hauber, 2010; Campbell & Hauber, 2009; D. L. Campbell & M. E. Hauber, 2010)). The key question is why females spend more time there. In such an experiment, the female is isolated from the group (or mate), which is likely stressful, and hence the experiment may not test mating preferences, but rather the female’s attempt to find her way back to familiar company. Also, the observed preference for the father is clearly a social preference and not an incestuous pairing preference. (B) In an operant conditioning test, isolated females can trigger the playback of sounds by pecking a key or hopping on a perch. These tests have yielded similar results as those in (A), but often with slightly weaker preference strength (rarely exceeding 60-65%; (Braaten & Reynolds, 1999; Hernandez, Perez, Mulard, Mathevon, & Vignal, 2016; Holveck & Riebel, 2014; Honarmand, Riebel, & Naguib, 2015; Riebel, 2000; Riebel, Naguib, & Gil, 2009)). This approach has the same problem regarding interpretation (social or mating preference). Interestingly, a recent study aimed at generating artificial dialects in zebra finches (Le Maguer, Derégnaucourt, & Geberzahn, 2021), carried out an operant conditioning test and found a weak preference for the natal dialect (about 55%). (C) Some early studies (N. S. Clayton, 1990; Nicky S Clayton & Pröve, 1989) claimed that female zebra finches express sexual preferences (“copulation solicitation display”) in response to playback of conspecific song after treatment with estradiol. However, it appears that Clayton may have misinterpreted the zebra finches’ response to stress (a horizontal sitting position) as solicitation of copulation (“crouching”). We also note that the observer (Clayton) was not blind to the treatment, and that the results

appear unrealistic (lack of variance within treatment groups). Later studies pointed out that true copulation solicitations (“tail quivering”) are exceedingly rare in such experimental circumstances, and the detailed analysis of ethograms predominantly leads to null results (Lin, Vanier, & London, 2014; Svec & Wade, 2009; Vyas, Harding, Borg, & Bogdan, 2009; Vyas, Harding, McGowan, Snare, & Bogdan, 2008).

Given the above, assume that we would carry out a song playback experiment and demonstrate that females preferentially approach the speaker playing unfamiliar songs of the own dialect rather than unfamiliar songs of an unfamiliar dialect. As the two treatments do not differ strongly in familiarity, the effect size would likely be small (see (Le Maguer et al., 2021)). However, independent of the strength of the effect, such an effect of dialect (variable X1) on the time spent close to a speaker (variable Z), simply cannot prove that dialect (X1) has a strong effect on pairing patterns (variable Y).

- (2) Our cross-fostering experiment was carried out with the explicit aim to disentangle culturally inherited traits from genetically inherited traits. In such an experiment, however, the binary predictor variable “rearing environment” encompasses all culturally inherited traits. It is hence unclear whether, in Table 2, the significant effect of the binary predictor labeled “Imprinting on song” is indeed a matter of dialect or of other culturally inherited traits (lines 203-205). To clarify this, we quantified how similar the song of a focal male is to the songs the focal female encountered in her rearing environment (a more specific version of the hypothesis). We quantified this in two ways (“ML song similarity to peers” and “SAP song similarity to peers”), and show that both these predictors simultaneously and significantly predict male-female distances, even while also accounting for “Imprinting on song” as a binary predictor. The effect of the variable “Imprinting on song” could potentially be explained by a confounding cultural trait such as foraging behavior, but this is difficult to imagine for the specific predictors of ML and SAP song similarity. Note that the observed variation in song similarity *within* dialects is largely a matter of coincidence (and not of a general sharing of culture *within* populations), because the focal males have never met any of the peers or parents from the female’s rearing aviary. It is hence impossible that other confounding cultural traits have been directly co-transmitted (lines 231-239). To “spuriously induce” the three correlations presented in Table 2 ($p=0.004$, $p=0.008$, $p=0.004$), any hypothetical confounding cultural variable would need to be so tightly linked to song characteristics that it would seem fair to describe it under the label of “song dialects” (lines 280-284). What else than a key component of song could cause each of these three variables to be significantly associated with male-female distances in a multivariate analysis? Note that the correlation between “ML song similarity to peers” and “SAP song similarity to peers” is only $r=0.17$. Hence, these variables can be considered two independent pieces of evidence, independent of the binary predictor “Imprinting on song”.

Please, note that it is not possible to produce direct experimental evidence that song dialects (X1) drive the pairing patterns (Y), because this would require experimental control over song. Although one could imagine that this can be achieved with robotic birds, females would unlikely form social pair bonds with such robotic birds. In contrast, a playback experiment can unequivocally show that females are capable of distinguishing the dialects (see also (Le Maguer et al., 2021)), but it cannot prove causality regarding the pairing patterns, as argued above. We therefore believe that the best possible evidence comes from a cross-fostering experiment in combination with observational data

showing that song similarity *per se* can predict the proximity of partners (Table 2). We are not aware of any other study that has provided stronger evidence that dialects drive assortment.

Reviewer #1 (Remarks to the Author):

In this study, the authors conducted a transgenerational experiment to test whether subtle differences in song between captive populations could persist across generations, and whether these differences also affect mating behaviour. The authors used four populations (two domesticated and two wild-derived) and ran a cross fostering experiment to disentangle culturally inherited traits from genetically inherited traits. The authors then used a novel machine learning methodology to show that there were subtle but significant differences between the songs of the populations, and that these differences persisted across generations. Finally, the authors showed that the song differences were associated with affiliative behaviour of females, such that females preferred to associate with males that sang songs that were similar to songs of the males from the rearing colony of the females.

This paper describes an interesting and novel study, and it is well written. There are, however, some issues that need to be addressed before the paper is ready to be published. Provided that these changes are made, I look forward to seeing this paper published in the future. I have explained my concerns below, but I also agree with some of the issues that were previously raised by the other reviewers.

Reply: We thank the reviewer for this positive assessment of our study.

Comments for the authors

Even though your results do not seem to be explained by song alone, you only briefly mention that you designed two post-hoc hypotheses to better explain the observed data. These hypotheses are interesting, but you do not discuss them sufficiently. You show a strong effect of rearing environment (which you assume to reflect only song learning) but that there is also an effect of phenotype. My point is that this could be due to the size of the birds, but it could also be other physiological traits that you did not measure. For example, there are sometimes clear deviations from wild-type coloration in domesticated zebra finches, and domesticated male zebra finches have higher testosterone levels compared to their wild counterparts (see Prior et al 2017 Gen Comp Endocrinol). I am not suggesting that you should measure these traits, but I would like to see some discussion of what other physiological traits may explain your results. This is particularly important considering that body size did not appear to explain your data very well (eg lines 211-212). On line 128 you mention that there may be other morphological traits, but this is not enough of a discussion. You also discuss your post-hoc hypotheses in the supplementary materials, but the manuscript would benefit from some of this discussion being incorporated in the main text.

Reply: Please note that, in response to advice from reviewers, we previously moved post-hoc explanations to the Supplement. We have now added some discussion about imprinting on morphotypes to the main text (lines 285-293), and we also added the information about testosterone levels (Prior et al. 2017) in the context of behavioural differences (see Supplementary Text).

There are potentially many other behavioural traits that zebra finches inherit from their rearing parent, and some of these traits could affect social dynamics. The manuscript would benefit from a more detailed discussion of these traits. For example, song rate is affected by the developmental environment in zebra finches (Tschirren 2009 *J Evol Biol*). One of your reasons for not discussing these possibilities (judging by your reply to previous comments) is that you can't achieve complete experimental control of song in this experiment. However, you could relatively easily test female preference experimentally through a playback experiment. Directly testing female song preference may be beyond the scope of this study, but you still need discuss the limitations of your paper and alternative mechanisms that could explain your results.

Reply: We now discuss possible confounding factors more specifically (see lines 203-205, 231-239, 280-284). As explained above (general reply), we see little scope for a confounding factor that could cause three significant correlations at once. Song rate is not a likely confound, as it does not seem to differ between domesticated and wild-derived populations (Tschirren, Rutstein, Postma, Mariette, & Griffith, 2009), and when creating selection lines that show only little overlap in song rate we found no tendency for assortative mating by selection line (D. Wang, Forstmeier, Martin, Wilson, & Kempenaers, 2020).

I agree with previous reviewers that the use of "cryptic dialects" is problematic. You have not defined this term, and I do not understand the reason why you use it. "Subtle population differences in song" would be easier for a reader to intuitively understand. If you can find a good scientific justification for the term (and explain it in the manuscript), I have no issue with it being used.

Reply: In principle we could replace "cryptic dialects" with "population differences in song that have remained hidden to researchers". However, for the title (and numerous other places) we prefer the short version. A "dialect" is simply a population difference in song that is strong enough (e.g. $d > 2$) to allow a clear classification (e.g. Figure 6a). We now define the use of the term dialect in line 317. We like the term "cryptic" to emphasize that the difference has remained unnoticed despite much research on the species (lines 313-314). In contrast, the term "subtle" seems problematic, because in many publications "subtle" is used to refer to differences of small effect size. Using "subtle" could be misleading the reader into thinking that the populations are largely overlapping (as previous studies concluded). We now avoid this term (lines 47, 312)

Minor comments

I had not heard of zebra finches not having dialects because of their variable song. This is an interesting hypothesis, but I am not entirely convinced. The zebra finch does not have a song that is particularly complex compared to species with larger repertoires. Also, some signs of dialects have already been described in this species (you cite this research). I suggest changing how you discuss the novelty of your results.

We edited the text (e.g. lines 42-43, 73, 298) to clarify that it is the individual distinctness of songs that has been considered a hindrance for the emergence of dialects. Only when there is a high consistency among males within a dialect, it is easy to characterise a dialect and to tell it apart from others.

Previous studies have described relatively small differences (in terms of effect sizes), and the most comprehensive and most recent study has declared these differences to be negligible and unlikely salient to the birds (Lachlan, van Heijningen, ter Haar, & ten Cate, 2016). Hence, we feel that it is fair to conclude that our findings are novel and unexpected.

You kept these birds in groups, meaning that the birds could learn open-ended and close-ended behavioural traits from several individuals. Zebra finches preferentially learn song from their social father, but they also learn song from other members of the colony. Did you gather any data on which potential tutors the cross fostered males associated with?

We did not gather data on associations between individuals during the rearing period. We could examine the song similarities to the foster father versus to other males, but learning from other males would not change the interpretation of this study.

How did you determine mate choice in the breeding experiment? Was it based on pair-bonding behaviour, barcode tracking, or both? Since you housed the birds in aviaries and you could observe them, it should have been easy to look at pair-bonding behaviour as well as mating behaviour (nest building, provisioning, etc). For example, how many of the 60 pairs from the first breeding experiment produced (genetic or EPC) offspring? If you are drawing conclusions about mate choice, this information would be highly relevant to this paper.

In the breeding experiments we observed pair-bonding and nesting behaviours as described in the Methods, but we did not conduct parentage analysis on the eggs that were laid. From a follow-up study (Forstmeier, Wang, Martin, & Kempenaers, 2021) we can confirm that the dialect preferences extend to extra-pair mating preferences (in the third generation).

Line 76 “acoustic space” is not a particularly intuitive term. This sentence would be easier to understand if it read something like: “In such cases, the need for individual recognition and distinctiveness may favour individuals that produce innovative or unusual songs, thereby eliminating the potential for local dialects.”

We thank the reviewer for this helpful suggestion. We changed the wording accordingly (Line 73).

81-84 I do not agree that this is the “prevailing view”. By doing a rather quick search, I was able to find at least one other study that used zebra finches to create salient (albeit artificial) “dialects” in captive populations. (Le Maguer 2021 Ethology). The results from Zann (1993) suggests that there are some differences in song between wild populations. These differences may be slight, but they were shown using traditional methods. It is interesting that you cite Lachlan here, considering that this paper did find differences between population, but they did not want to draw conclusions about the signal’s salience. It would be more accurate to say that only subtle population have been found in this species, and that it is unclear whether these differences are salient. I suggest that you cite Le Maguer here (and where it suits).

While we are happy to cite the new study by (Le Maguer et al., 2021) (line 84, 323), it is not comparable to ours. Importantly, it does not describe the natural situation in which each male sings its individual-specific song (no conformity). Instead, the study attempts to create artificial dialects with high conformity by putting together many males that all had the same tutor. This study also does not question the prevailing view that we describe in our manuscript.

The prevailing view is clearly explained in (Lachlan et al., 2016), and their view was also adopted in an authoritative review (Tchernichovski, Feher, Fimiarz, & Conley, 2017). Lachlan et al. (2016) declared the population differences to be negligible, and they in fact did draw conclusions about the signal’s salience, as we quoted in line 103 of the previous version of the manuscript (now line 94).

128 If you are mentioning “unmeasured aspects of phenotype” you should also mention that there are possible culturally inherited traits that you have not measured.

We agree. At this place in the manuscript, it seems more appropriate to only mention the traits that we specifically measured, and to mention unmeasured traits in the discussion. Hence, we removed the “and unmeasured aspects of the morphotype” (line 130).

191-196 I do not understand why the best explanation for the observed data has been moved to the supplementary materials. You should include this in the main text.

This was done in response to previous reviewers’ comments. The change to move post-hoc explanations to the Supplement was perceived positively. To find a compromise, we have now added some more discussion to the main text (lines 285-293). We would also be willing to move the post-hoc explanations back to the main text, if the editor feels that this is an improvement.

224-225 You do not have to achieve full experimental control over songs to experimentally test female preference. For example, a playback experiment could determine if the females can discriminate between these dialects.

As explained above (general reply), we feel that such an experiment cannot really help with establishing causality, whereas the analyses of song similarity values present strong evidence.

Line 228-241. If they learn other aspects of behaviour than just song (such as foraging patterns or song rate), these traits could be passed on to the subsequent generation just as much as song. I would recommend using this paragraph to discuss your post hoc hypotheses. It is interesting that you found an effect of domesticated phenotype in addition to song, and this is something that should be discussed in the main text. Alternatively, you could use this paragraph to discuss what other traits may be culturally inherited in this species, and the underlying mechanisms.

We thank the reviewer for this helpful comment. We now explain better that other culturally inherited traits could potentially induce the effect of the binary variable “Imprinting on song” in Table 2 (lines 203-205). However, such traits cannot induce spurious effects of the two song similarity measures, because similarity to the peers is largely coincidental. We further explain that these birds have not met and have not been reared by the same parents, and hence they cannot have exchanged other cultural traits in proportion to song similarity (lines 231-239). Please, further note that one would expect that culturally inherited foraging patterns also induce a population separation at the same-sex level, but this is not the case (Fig. 5b). Birds also do not mate assortatively for song rate (D. Wang et al., 2020).

259 I disagree with this statement. Your study shows that culturally inherited differences between populations affect assortative mating behaviour in zebra finches. You do not directly test if song is the mechanism for this assortative mating.

We respectfully disagree. Table 2 presents two independent tests (SAP and ML), and both are significant simultaneously. They both provide strong support that peers are more influential than parents.

263 Again, it seems to me that most papers on this topic use a much more cautious language when discussing dialects in zebra finches. When you use phrases such as “the entire space of acoustic and syntactical possibilities” that indicates that there are no dialects, but the papers you cite have found small differences between populations.

We feel that it is important to discuss this in terms of effect sizes. The conventional methods of quantifying differences have yielded small effects (see e.g. Figure 6c,d), while the machine-learning

approach shows essentially non-overlapping distributions (see Figure 6a,b). In response to this comment, we rephrased the sentence (line 298).

Line 266: Differences in song can be either cryptic or striking, they cannot be both. Please change the wording here, and carefully look over the manuscript to make sure the wording is correct and consistent.

We rephrased the sentence (lines 303-305). Striking referred to the large effect sizes (see Figure 6a, b), while “cryptic” refers to the fact that this difference has remained unnoticed for so long (due to methodological limitations; lines 313-314).

268 Add a citation to this statement.

The citation was in the following line, but we agree that it should be added here as well (line 306).

271 Do you mean that zebra finches belong to this group? I would argue that many songbirds have a much more complex song than the zebra finch. Zebra finches may have a variable song (large differences between individuals), but that is not the same as having a complex song (long and variable song motifs or a large repertoire).

Indeed, the zebra finch song is not complex, but highly individualistic. We try to make clear throughout the manuscript (see additional edits in lines 42-43, 73, 298). Please note that the conclusion that there may be cryptic dialects despite limited conformity applies to both cases (high variability within and between males).

276-279 Please find a better motivation for the use of “cryptic dialects”. Previous studies have found subtle song differences between populations, and your machine learning approach also finds subtle (but salient) differences between populations. What justifies “cryptic dialect” in this case? Do dialects have to be salient to females according to your definition?

As explained above, we feel that the difference in interpretation is justified by the large difference in effect sizes (see Fig. 6). We now refer to this figure to emphasize the large effect (lines 304, 317) and we avoid using the term ‘subtle’ to not imply that the distributions may be largely overlapping (lines 47, 312).

Line 291: I do not agree that it is arguably closer to the biological reality. Often machine learning approaches are sensitive to small differences, but they are also a black box. In order to claim that a method is “close to biological reality” it would need to take into account the sensory biology of the animal model. A machine learning approach would easily find differences that cannot be perceived by the animal and there would be no way for the researcher to know this without using a behavioral approach.

We doubt that machine learning algorithms would be superior to animals in distinguishing classes of stimuli, as long as we consider stimuli that (1) match the sensory capacities of the animal and that (2) have great biological relevance to the animal. Animal brains can achieve a lot in the domains they have been extensively trained in (e.g. abilities in individual recognition are quite astonishing). We essentially argue that machine learning algorithms may be able to achieve similar performance as the trained animal brain, while conventional approaches may not.

598-604 I have not heard of using SAP like this. My guess is that you only looked at overall similarity? How does these “super motifs” compare to a single song in terms of sequential similarity, accuracy, etc.? Did you work directly with the developers of SAP when designing

We took this approach – without asking Ofer Tchernichovski for advice – because male zebra finches often learn from multiple tutors. Comparing the tutees’ song to each of the tutors separately will then not produce a close match with any of them. In contrast, our approach with the ‘super-motif’ allows all elements of the tutees’ song to be matched to the entire repertoire of all tutors at once. We indeed focussed on overall similarity, because e.g. sequential similarity would be difficult to interpret in relation to such a composite recording.

Reviewer #2 (Remarks to the Author):

I was Reviewer 1 during the review process for Nature Ecology and Evolution. The aim of the study is to explore whether the songs of four captive populations of zebra finches have subtle, but salient differences to females. The authors conclude that there are subtle song differences among populations (which has been long known, as stated by the authors in lines 86-89) and that these differences guide female mate choice (which is the main novel claim of the paper). Carrying over from my previous review, I have issues with both of these conclusions. This was a hard review to write, because I think the authors’ findings are really interesting. However, their conclusions go way beyond the data they actually have. Without an experiment isolating the influence of other environmental factors, I don't think that the authors can claim song differences determine mate choice.

As we explained above (general reply), we respectfully disagree with the reviewer. In Table 2, the binary predictor “Imprinting on song” statistically controls for all population-level confounding traits that are culturally inherited, whereas the two significant individual-level predictors of song similarity cannot be confounded by co-inherited cultural traits because the focal males and the peers from the natal aviary have never met and were not reared by the same parents. Hence, our experimental design controls for such potential confounds. This has now been clarified in the main text.

1) I was surprised and somewhat troubled to read the authors’ statements that it is not necessary to define what a dialect is and that defining a dialect is now “outdated” given machine learning approaches. There are two issues here.

First, there is a serious intellectual issue: the authors are bold enough to use the term “cryptic song dialect” to, as they write “attract the interest of potential readers”, but not willing to actually state what they mean by the terms “dialect” or “cryptic”.

We acknowledge that the previous stance was a bit too strongly stated. We use the term “dialect” to refer to a population difference in song that is strong enough (e.g. effect size $d > 2$) to allow a clear classification (Figure 6a). Whether one draws a line at a classification success of 80% or 99% can be debated, but a tiny population difference (e.g. $d = 0.1$ and 55% success) would not be sufficient. This is now explained in line 317. We use the term “cryptic” because this population difference has remained unnoticed despite much research on the species (line 314).

Second, the authors themselves seem confused about what a dialect is: On the one hand, they claim that machine learning is sufficient to prove dialects, which seems to be implicitly arguing that any detectable difference between any two groups constitutes a dialect. In this case, as I wrote in the previous round of reviews, it is likely that machine learning approaches would demonstrate differences on many types of arbitrary groupings of individuals. What is the null hypothesis for cryptic dialects? Without this, it is impossible in my view to conclude whether the authors’ results are at all surprising. (And they are indeed not, since earlier work has already shown some geographic

differences in zebra finch songs). On the other hand, the authors say that any song variation associated with response variation constitutes a dialect. But we know that song variation within a population exists and has behavioral effects—are we now supposed to say that there are within-population and within-family group dialects as long as the songs from one group are preferred over those from another?

A population difference in song is traditionally called a “dialect”, while a family difference would be called something else (e.g. “family-specific contact calls”, (Sharp & Hatchwell, 2006)). Our goal is not to re-define the term dialect. We just use the term that is most commonly used for discussing population differences in song. We could replace the term “dialect” with “population difference in song”, but then readers would probably wonder why we avoid using the term dialect.

The study by Lachlan et al. (2016) and adopted in the review by Tchernichovski et al. (2017) concludes (1) that population differences in song are negligible and (2) that these differences are likely of no relevance to the birds. Our study shows evidence that clearly suggests that both of these conclusions are wrong.

2) I was also surprised to read the authors’ statement that an experiment to test whether song preferences guide mate choice is not possible. What about testing female discrimination and/or preferences for songs? This would clearly be able to isolate song as a factor influencing mate choice. I realize that such experiments would obviously be some work to complete (and maybe not possible given the state of the colonies), but there are clearly options aside from constructing robot birds!

As we explained above (general reply), playback experiments are unable to establish causality. It is often stated that playback experiments (or choice tests in general) have been validated in the sense that they reflect mating preferences, but such validation would be needed for each specific setup (here for the study of dialects) and effect sizes would need to be high. For instance, male courtship rate (X) significantly predicted the amount of time females spent next to males in a choice chamber (Y), and Y significantly predicted female responsiveness (Z) in extra-pair mating trials (Forstmeier 2007). Nevertheless, the correlation between X and Z is zero (Forstmeier, 2007). Hence, extrapolation does not work, despite validation of the assay (in terms of Y predicting Z). In light of this, and of the low repeatability of preferences measured in choice chamber trials (Forstmeier & Birkhead, 2004), we decided against running a playback experiment. Instead, we added the work on generation 3, which we find more convincing.

3) Although the authors feel that they have toned down their conclusions, statements such as “Our study shows that population-specific song dialects drive strong assortative mating in zebra finches” and “we reveal experimentally that these ‘cryptic song dialects’ have real consequences for social behaviour” and “we show that zebra finches are surprisingly sensitive to population differences in song during the process of mate choice” are not at all cautious. The manuscript repeatedly equates culture with song e.g., “Hypothesis 3 assumes a learnt preference for a cultural trait, such that all birds prefer to mate with a partner from their own cultural population because of socially transmitted variation in song characteristics.” Without a proper experiment showing, making this conclusion depends on eliminating any reasonable alternative explanation. However, I am unconvinced by the authors’ arguments that alternative learned or inherited behavioral traits could not play a critical role in mate choice. The authors’ main argument against this is that they know that song has a fixed sensitive period and then they assume that any other behavioral trait has an open-ended learning period (line 232) and would be mostly learned from the social group rather than the parents. Aside from the lack of any evidence that other learned traits are open-ended, this argument could not eliminate other inherited effects that might be correlated with song, such as early stress.

We agree with the reviewer that other culturally inherited traits could potentially have induced the effect of the binary variable “Imprinting on song” (Table 2). However, these traits cannot have induced spurious effects of the two song similarity measures (Table 2), because the birds involved have not met and have not been reared by the same parents, and hence they cannot have exchanged other cultural traits in proportion to song similarity (see also the general reply to reviewers 1, 2, 4). We now try to explain this better and thereby better justify our conclusions (lines 203-205, 231-239, 280-284).

However, given that I brought it up in a previous review, I’ll specifically discuss distance calls being incorporated into the song. Distance calls are included in the song (in fact often the loudest element), are learned from the parents, and expresses variation both geographically as well as between lab and wild populations. Moreover, the acoustic structure of the distance call strongly influences social-behavioral responses (Vignal and Mathevon 2011 in J Comp Psych). Why couldn’t then the inherited distance call, which is also found in the song, play a key role in social interactions and mate choice?

The authors’ main argument here is an earlier study showing a weak acoustic correlation between a male’s own distance call and his song. There are a few issues with this argument. First, this doesn’t at all address the question of whether the distance call in the song is itself similar to the distance calls heard early in life, which is the relevant comparison.

The reviewer’s statement about the “relevant comparison” is incorrect. We show that both “ML similarity to peers” (X1) and “SAP similarity to peers” (X2) significantly predict male-female distances (Y; Table 2). Previously, the reviewer argued that these relationships could be spurious and induced by a confounding variable (X3), namely the similarity of the focal male’s distance call to the distance calls in the female’s natal aviary (i.e. females choose males based on their distance calls rather than their songs). X3 can only induce spurious correlations between X1 and Y (and between X2 and Y) if X3 is strongly correlated with X1 (and Y, and X2). A strong correlation with X1 is only possible if a male’s distance call is similar to its own song. We have previously shown that a male’s song and its own distance call are only weakly correlated (Forstmeier, Burger, Temnow, & Derégnaucourt, 2009), hence X1 and X3 cannot be strongly correlated. Therefore, the contact call cannot induce a spurious correlation between song measures and pairing. We agree that it is possible that the distance call plays an *additional* role that we did not capture. However, the issue at hand is whether calls may be confounding, rather than possibly playing an independent, additional role. Such an independent role seems somewhat unlikely because distance calls are not used in male-female courtship but appear to function in reuniting a pair that has lost contact.

Other questions about the distance call can of course be asked, but we do not see how this is relevant for interpreting the results shown in Table 2. Call elements that are part of the song of the focal male are comprised in our ML and SAP similarity measures, but similarity is only quantified relative to the songs of the birds in the female’s rearing aviary, and not to their calls. In this sense, calls could play an additional role, and our measures of similarity would then be suboptimal (but apparently still good enough to predict male-female distances).

Second, I don’t see how the correlation between a signal element and the whole song is appropriate. Certain elements within or associated with songs may play a key role in responses, rather than the overall acoustic characteristics of the song. For example, the whistle in white-crowned sparrow songs or the chatter call in cowbird songs. The whistle alone might have a low acoustic correlation with an entire white-crowned sparrow song, but this does not mean the whistle is not found in the song. The relevant comparison here would be to identify songs having the distance call as

introductory or concluding element and compare that with the distance calls of the early social environment.

We are puzzled by this comment. We have no data on distance calls and even if we did, it is not clear how the suggested approach would inform us about whether X3 could induce a spurious correlation between X1 and Y.

I think that the authors can conclude no more than that: 1) there are some differences in the songs of the groups they've studied and 2) there is some trait that is influenced by early life that is itself correlated with song (and might possibly be the song or some associated vocal trait) that influences mate choice.

For a hypothetical confounding cultural trait to induce three spurious correlations (Table 2), it would need to be tightly correlated with each of the three predictors. Given that the correlation between "ML similarity to peers" and "SAP similarity to peers" is $r=0.17$, it is mathematically impossible that a single confounding factor can achieve this. Or would we have to consider two such confounding factors? In that case, it is remarkable that both would induce the same biological conclusion, namely that the song of the peers is more influential than the song of the parents (Table 2). Moreover, how could two such confounds arise in the absence of a plausible mechanism, given that the focal males never met the peers of the female? In sum, we feel that our conclusions are justified.

Minor comments:

Abstract

Lines 55-56: Isn't this extremely well-known? Essentially the authors are saying that human observers haven't previously detected differences that are relevant to the birds themselves, but I think this is widely accepted.

We strongly disagree. If this had been widely known and accepted, why has it been ignored in experimental designs? And how does this fit with the conclusions of (Lachlan et al., 2016)?

Introduction

Overall, the introduction is very zebra finch focused—is this really a technical zebra finch-researcher question and study or of broader relevance? I think the authors could do more to broaden the appeal of this study.

As the extended discussion with the reviewers shows, it seems essential to explain the study system and the background knowledge in sufficient detail. Given space limitations, we prefer to highlight the broader relevance in the Discussion section (lines 294-337).

Lines 106-110: why is sexual imprinting contrasted with song learning? Can't song learning be understood as a kind of sexual imprinting?

We edited these two sentences (lines 115, 117) to clarify that the contrast is between sexual imprinting on size (largely genetically inherited) versus sexual imprinting on song (largely culturally inherited).

Line 110: why the term "cultural" drift? Couldn't there be environmental reasons? Couldn't there be variation in receivers that make certain song types more effective?

Cultural drift is the most parsimonious explanation for the divergence of songs (and of languages in humans). Adaptation of song traits to environments or receiver psychology should be negligible over maximally 100 generations, given that sexual selection is weak in this species due to a lack of

consensus among females (Forstmeier & Birkhead, 2004; Ihle, Kempnaers, & Forstmeier, 2015; D. Wang et al., 2020; D. P. Wang, Forstmeier, & Kempnaers, 2017).

Discussion

Lines 256-257: This is incredibly jargony. I honestly have no idea what is meant by “each zebra finch population covers the entire space of acoustic and syntactical possibilities”. Do the authors mean that every population, no matter how small or uniform produces all possible songs???

We rephrased this sentence to avoid jargon and clarify what we mean (line 298).

Line 260: throughout, the words “subtle” and “cryptic” are used, but now the differences are referred to as “striking”—based on what?

We now avoid using the word “striking” (line 304), but what we meant and referred to is the large effect size (with non-overlapping distributions, see Fig. 6a, b).

Line 262: is this a case of inadequate sampling or a true fact?

We added a citation to make clear that this is a true fact (line 306).

Line 283: “closer to the biological reality of an individual brain that distinguishes familiar from unfamiliar” What does this mean?

There are obvious parallels in discrimination learning between brains and artificial systems. Such parallels are not present in the mentioned alternative approaches. We deleted “the biological reality of” (line 332).

And how does machine learning do this better than actually testing what a female responds to?

As we explained, the former is easy to run and the latter difficult to study. We are not saying that it is better, just easier.

Wouldn't a discriminant analysis on a suite of song traits do the same?

Possibly yes. However, given the small effect sizes reported by Lachlan et al. 2016, the ability to classify songs might be limited, especially when considering that the data set would need to be split into a training set (for setting up the discriminant function) and a testing set for validation.

Line 287: I would add that it might also pick up non-relevant differences in the songs due to background noise, recording techniques, etc... etc...

Changed as suggested (Lines 336-337).

Reviewer #3 (Remarks to the Author):

Overall, the authors appear to have taken good coherent actions to respond to the reviewers.

In this round, I again concentrate primarily on the machine learning aspect.

The authors have correctly addressed one of my main concerns (the possibility of confounding acoustic factors). I now have no objection about that.

My other main concern is the opacity of the ML used (Apple's "CreateML") which creates a risk against reproducibility of the results, or the ability of future researchers to apply the same approach even when this software version is changed/gone. The researchers have provided software version information, which is helpful; and with the publication of the audio data (as pledged) and associated information, I am happy to accept the claim that adding another ML classifier would overcomplicate the paper. I remain concerned that this aspect is not conducive to open science, and I hope that in future the analysis can be performed using a fully open software approach.

Thus, I am content to recommend the paper be accepted.

We thank the reviewer for the positive assessment.

Reviewer #4 (Remarks to the Author):

I have reviewed this manuscript before (as reviewer 2) and some main issues remain:

1. There is no direct test for the effect of song, per se, on mate choice. Table 2 reveals a correlation but cannot rule out effects of other socially inherited traits (there may be many other "cryptic" traits).

We now explain better that the song similarity measures in Table 2 cannot easily be confounded by other socially inherited traits, given that the focal males and the female's peers never met and also were not raised by the same parents (lines 231-239, 280-284).

2. It is unclear when song is important during the pairing process. (Information lacking about the pairing process)

We are puzzled by this comment. What kind of information could we have provided that is not already presented in Figure 5? The pairing process may take several days or even weeks until a pair bond is strong enough to make divorce unlikely, so it is hard to see how data on numerous short episodes of singing by the future partner and other males (which we do not have) could be linked to data on pairwise distances.

3. Pairing patterns are monitored in large aviaries meaning that the mate choice of any particular bird is contained by the mate choice made by other individuals. How is this problem solved by the analyses?

This problem is inherent to the socially monogamous system that we study, and we are not aware of techniques that would solve the problem during data analysis. Pair bonds are established gradually, so depletion of partners (in terms of reduced time available) already starts with the first encounters of mates. We think that this problem cannot be solved in the analyses, but we agree with the reviewer that this is an interesting issue that deserves further study.

References

- Braaten, R. F., & Reynolds, K. (1999). Auditory preference for conspecific song in isolation-reared zebra finches. *Animal Behaviour*, *58*(1), 105-111.
- Campbell, D., & Hauber, M. (2010). Behavioural correlates of female zebra finch (*Taeniopygia guttata*) responses to multimodal species recognition cues. *Ethology Ecology & Evolution*, *22*(2), 167-181.
- Campbell, D. L., & Hauber, M. E. (2009). Cross-fostering diminishes song discrimination in zebra finches (*Taeniopygia guttata*). *Animal cognition*, *12*(3), 481-490.
- Campbell, D. L., & Hauber, M. E. (2010). Conspecific-only experience during development reduces the strength of heterospecific song discrimination in zebra finches (*Taeniopygia guttata*): a test of the optimal acceptance threshold hypothesis. *Journal of Ornithology*, *151*(2), 379-389.
- Clayton, N. (1988). Song discrimination learning in zebra finches. *Animal Behaviour*, *36*(4), 1016-1024.
- Clayton, N. S. (1990). Assortative Mating in Zebra Finch Subspecies, *Taeniopygia-Guttata-Guttata* and *T-G-Castanotis*. *Philosophical Transactions of the Royal Society of London Series B-Biological Sciences*, *330*(1258), 351-370. doi: DOI 10.1098/rstb.1990.0205
- Clayton, N. S., & Pröve, E. (1989). Song discrimination in female zebra finches and Bengalese finches. *Animal Behaviour*.
- Forstmeier, W. (2007). Do individual females differ intrinsically in their propensity to engage in extra-pair copulations? *PLoS one*, *2*(9), e952.
- Forstmeier, W., & Birkhead, T. R. (2004). Repeatability of mate choice in the zebra finch: consistency within and between females. *Animal Behaviour*, *68*(5), 1017-1028.
- Forstmeier, W., Burger, C., Temnow, K., & Derégnaucourt, S. (2009). The genetic basis of zebra finch vocalizations. *Evolution: International Journal of Organic Evolution*, *63*(8), 2114-2130.
- Forstmeier, W., Wang, D., Martin, K., & Kempenaers, B. (2021). Fitness costs of female choosiness are low in a socially monogamous songbird. *PLoS biology*, *19*(11), e3001257.
- Hernandez, A. M., Perez, E. C., Mulard, H., Mathevon, N., & Vignal, C. (2016). Mate call as reward: Acoustic communication signals can acquire positive reinforcing values during adulthood in female zebra finches (*Taeniopygia guttata*). *Journal of Comparative Psychology*, *130*(1), 36.
- Holveck, M.-J., & Riebel, K. (2014). Female zebra finches learn to prefer more than one song and from more than one tutor. *Animal Behaviour*, *88*, 125-135.
- Honarmand, M., Riebel, K., & Naguib, M. (2015). Nutrition and peer group composition in early adolescence: impacts on male song and female preference in zebra finches. *Animal Behaviour*, *107*, 147-158.
- Ihle, M., Kempenaers, B., & Forstmeier, W. (2015). Fitness Benefits of Mate Choice for Compatibility in a Socially Monogamous Species. *Plos Biology*, *13*(9). doi: ARTN e100224810.1371/journal.pbio.1002248
- Lachlan, R. F., van Heijningen, C. A. A., ter Haar, S. M., & ten Cate, C. (2016). Zebra Finch Song Phonology and Syntactical Structure across Populations and Continents-A Computational Comparison. *Frontiers in Psychology*, *7*. doi: ARTN 98010.3389/fpsyg.2016.00980
- Le Maguer, L., Derégnaucourt, S., & Geberzahn, N. (2021). Female preference for artificial song dialects in the zebra finch (*Taeniopygia guttata*). *Ethology*, *127*(7), 537-549.
- Lin, L. C., Vanier, D. R., & London, S. E. (2014). Social information embedded in vocalizations induces neurogenomic and behavioral responses. *PLoS One*, *9*(11), e112905.
- Miller, D. B. (1979a). The acoustic basis of mate recognition by female zebra finches (*Taeniopygia guttata*). *Animal Behaviour*, *27*, 376-380.
- Miller, D. B. (1979b). Long-term recognition of father's song by female zebra finches. *Nature*, *280*(5721), 389-391.

- Riebel, K. (2000). Early exposure leads to repeatable preferences for male song in female zebra finches. *Proceedings of the Royal Society of London. Series B: Biological Sciences*, 267(1461), 2553-2558.
- Riebel, K., Naguib, M., & Gil, D. (2009). Experimental manipulation of the rearing environment influences adult female zebra finch song preferences. *Animal Behaviour*, 78(6), 1397-1404.
- Sharp, S. P., & Hatchwell, B. J. (2006). Development of family specific contact calls in the Long-tailed Tit *Aegithalos caudatus*. *Ibis*, 148(4), 649-656.
- Svec, L. A., & Wade, J. (2009). Estradiol induces region-specific inhibition of ZENK but does not affect the behavioral preference for tutored song in adult female zebra finches. *Behavioural brain research*, 199(2), 298-306.
- Tchernichovski, O., Feher, O., Fimiarez, D., & Conley, D. (2017). How social learning adds up to a culture: from birdsong to human public opinion. *Journal of experimental biology*, 220(1), 124-132.
- Tschirren, B., Rutstein, A., Postma, E., Mariette, M., & Griffith, S. (2009). Short- and long-term consequences of early developmental conditions: a case study on wild and domesticated zebra finches. *Journal of evolutionary biology*, 22(2), 387-395.
- Vyas, A., Harding, C., Borg, L., & Bogdan, D. (2009). Acoustic characteristics, early experience, and endocrine status interact to modulate female zebra finches' behavioral responses to songs. *Hormones and behavior*, 55(1), 50-59.
- Vyas, A., Harding, C., McGowan, J., Snare, R., & Bogdan, D. (2008). Noradrenergic neurotoxin, N-(2-chloroethyl)-N-ethyl-2-bromobenzylamine hydrochloride (DSP-4), treatment eliminates estrogenic effects on song responsiveness in female zebra finches (*Taeniopygia guttata*). *Behavioral neuroscience*, 122(5), 1148.
- Wang, D., Forstmeier, W., Martin, K., Wilson, A., & Kempenaers, B. (2020). The role of genetic constraints and social environment in explaining female extra-pair mating. *Evolution*, 74(3), 544-558.
- Wang, D. P., Forstmeier, W., & Kempenaers, B. (2017). No mutual mate choice for quality in zebra finches: Time to question a widely held assumption. *Evolution*, 71(11), 2661-2676. doi: 10.1111/evo.13341
- Woolley, S. C., & Doupe, A. J. (2008). Social context-induced song variation affects female behavior and gene expression. *PLoS biology*, 6(3), e62.
- Zann, R. (1993). Variation in song structure within and among populations of Australian zebra finches. *The Auk*, 110(4), 716-726.

Reviewers' Comments:

Reviewer #1:

Remarks to the Author:

Most of my comments have been addressed in the response to reviewers and/or manuscript. There are obviously some comments that the authors strongly disagree with, and I appreciate that the authors give an explanation as to why. I have no more comments on this manuscript.

I look forward to seeing the final version of the manuscript.

Reviewer #2:

Remarks to the Author:

This is now the third time I've reviewed this paper and I will keep this short, since none of my previous issues with the study were addressed in this revision. Essentially, I leave it up to the editor, since my recommendation does not change.

1) Even if a reader would have no issues with the manuscript and completely accepts the authors' claims, the implications of the study are incredibly zebra finch specific. Essentially, the only, as far as I can see, consequence or novelty of the paper is overturning a claim in a 2016 paper on song phonology in zebra finches. The existence of dialects in other species (and the consequences of dialect differences alone for mate choice) have been known for decades.

2) The authors use "cryptic" and "dialect" without defining these terms. There is no null hypothesis for such a machine learning approach.

3) The authors argue that correlational evidence is more reliable than an experiment that isolates the effects of song, given previous challenges with designing such experiments. However, the critical point remains that, if rearing group influences song, it is incredibly likely to influence other socially acquired traits. Thus, an apparent effect of song might be only correlational.

Reviewer #4:

Remarks to the Author:

Because this is the third time I review this manuscript, I only give short comments to the responses given by the authors below:

1. There is no direct test for the effect of song, per se, on mate choice. Table 2 reveals a correlation but cannot rule out effects of other socially inherited traits (there may be many other "cryptic" traits). We now explain better that the song similarity measures in Table 2 cannot easily be confounded by other socially inherited traits, given that the focal males and the female's peers never met and also were not raised by the same parents (lines 231-239, 280-284).

Comment: These additions make the discussion more balanced. I especially appreciate that the authors clearly state that the evidence that song per se affect mate choice is correlational.

2. It is unclear when song is important during the pairing process. (Information lacking about the pairing process)

We are puzzled by this comment. What kind of information could we have provided that is not already presented in Figure 5? The pairing process may take several days or even weeks until a pair bond is strong enough to make divorce unlikely, so it is hard to see how data on numerous short episodes of singing by the future partner and other males (which we do not have) could be linked to data on

pairwise distances.

Comment: I asked this question because I have never observed how zebra finches use song in courtship and mate choice. In some bird species, males mainly use song to attract females from a long-distance. Figure 5 does not include a description of how and when song is used in this species. Adding this type of information would help to interpret the importance of the findings in a more natural setting.

3. Pairing patterns are monitored in large aviaries meaning that the mate choice of any particular bird is contained by the mate choice made by other individuals. How is this problem solved by the analyses?

This problem is inherent to the socially monogamous system that we study, and we are not aware of techniques that would solve the problem during data analysis. Pair bonds are established gradually, so depletion of partners (in terms of reduced time available) already starts with the first encounters of mates. We think that this problem cannot be solved in the analyses, but we agree with the reviewer that this is an interesting issue that deserves further study.

Comment: This is true. But the authors have data on daily movements of the birds meaning that they could actually extract information about the sequence of pairing events.

Reply to reviewer comments

Reviewer #1 (Remarks to the Author):

Most of my comments have been addressed in the response to reviewers and/or manuscript. There are obviously some comments that the authors strongly disagree with, and I appreciate that the authors give an explanation as to why. I have no more comments on this manuscript.

I look forward to seeing the final version of the manuscript.

Reply: We thank the reviewer for this comment.

Reviewer #2 (Remarks to the Author):

This is now the third time I've reviewed this paper and I will keep this short, since none of my previous issues with the study were addressed in this revision. Essentially, I leave it up to the editor, since my recommendation does not change.

1) Even if a reader would have no issues with the manuscript and completely accepts the authors' claims, the implications of the study are incredibly zebra finch specific. Essentially, the only, as far as I can see, consequence or novelty of the paper is overturning a claim in a 2016 paper on song phonology in zebra finches. The existence of dialects in other species (and the consequences of dialect differences alone for mate choice) have been known for decades.

Reply: We find this comment puzzling, but will leave it up to the readers to decide whether our study is interesting or presents novel results. We just note that the zebra finch has been a widely used model system for studies on song.

2) The authors use "cryptic" and "dialect" without defining these terms. There is no null hypothesis for such a machine learning approach.

Reply: We added a definition as requested by the Editor. As we stated earlier, "cryptic" just means that this has remained hidden to researchers, and "dialects" are population differences in songs.

We verified that the machine learning approach confirms the null hypothesis in cases where no differentiation between two groups exists (in our example, 49.5% correct assignments, with chance level being 50%). Hence, population differentiation is a gradual phenomenon, and outcomes can vary from 50% to 100% correct assignments. We think that this is a useful way of quantifying differentiation.

3) The authors argue that correlational evidence is more reliable than an experiment that isolates the effects of song, given previous challenges with designing such experiments. However, the critical point remains that, if rearing group influences song, it is incredibly likely to influence other socially acquired traits. Thus, an apparent effect of song might be only correlational.

Reply: The raised argument misses an important aspect of our analysis, namely that we examine variation within rearing aviaries rather than between rearing groups. After accounting for dialect as a binary trait, we show that the focal male's song similarity to the song of the peers in the female's

rearing aviary influence the female's choice. This shows that the preferences target song familiarity beyond the dichotomy of dialects. Because both measures of song similarity are simultaneously significant, the effect of song similarity must be rather large. Indeed, when the binary predictor of dialect is removed from the model, the machine-learning-based measure of song similarity takes up nearly all of its variance. Hence the measure explains the preference for the own dialect, and it explains variation in preferences even within populations.

We agree that the rearing group will also influence other socially acquired traits, but these traits cannot explain why the two predictors 'ML similarity to peers' and 'SAP similarity to peers' both significantly explain mate choice. A close co-inheritance between songs and other traits across multiple generations within populations is also highly unlikely, and if they exist, such closely related traits can be regarded as part of the song culture.

Reviewer #4 (Remarks to the Author):

Because this is the third time I review this manuscript, I only give short comments to the responses given by the authors below:

1. There is no direct test for the effect of song, per se, on mate choice. Table 2 reveals a correlation but cannot rule out effects of other socially inherited traits (there may be many other "cryptic" traits).

[taken from our previous reply]: We now explain better that the song similarity measures in Table 2 cannot easily be confounded by other socially inherited traits, given that the focal males and the female's peers never met and also were not raised by the same parents (lines 231-239, 280-284).

Comment: These additions make the discussion more balanced. I especially appreciate that the authors clearly state that the evidence that song per se affect mate choice is correlational.

Reply: We thank the reviewer for this comment.

2. It is unclear when song is important during the pairing process. (Information lacking about the pairing process)

[taken from our previous reply]: We are puzzled by this comment. What kind of information could we have provided that is not already presented in Figure 5? The pairing process may take several days or even weeks until a pair bond is strong enough to make divorce unlikely, so it is hard to see how data on numerous short episodes of singing by the future partner and other males (which we do not have) could be linked to data on pairwise distances.

Comment: I asked this question because I have never observed how zebra finches use song in courtship and mate choice. In some bird species, males mainly use song to attract females from a long-distance. Figure 5 does not include a description of how and when song is used in this species. Adding this type of information would help to interpret the importance of the findings in a more natural setting.

Reply: We now added a description to the Discussion section that the courtship song that we recorded for our analyses is directly used in the process of mate choice.

3. Pairing patterns are monitored in large aviaries meaning that the mate choice of any particular bird is contained by the mate choice made by other individuals. How is this problem solved by the analyses?

[taken from our previous reply]: This problem is inherent to the socially monogamous system that we study, and we are not aware of techniques that would solve the problem during data analysis. Pair bonds are established gradually, so depletion of partners (in terms of reduced time available) already starts with the first encounters of mates. We think that this problem cannot be solved in the analyses, but we agree with the reviewer that this is an interesting issue that deserves further study.

Comment: This is true. But the authors have data on daily movements of the birds meaning that they could actually extract information about the sequence of pairing events.

Reply: We did analyse the latency to form pair bonds, but the results were largely identical to those that are being presented, namely in terms of the number of pairs in each of the 16 categories (male-female combinations). Categories in which many pair bonds were formed were also those where pair bonds were formed early (suggesting preferred mating). Categories with few pair bonds showed delayed pair-bond formation (suggesting non-preferred mating).